# Seed choice in ground beetles is driven by surface-derived hydrocarbons

Khaldoun A. Ali [1✉], Boyd A. Mori[2], Sean M. Prager [1] & Christian J. Willenborg[1]

Ground beetles (Coleoptera: Carabidae) are among the most prevalent biological agents in temperate agroecosystems. Numerous species function as omnivorous predators, feeding on both pests and weed seeds, yet the sensory ecology of seed perception in omnivorous carabids remains poorly understood. Here, we explore the sensory mechanisms of seed detection and discrimination in four species of omnivorous carabids: *Poecilus corvus*, *Pterostichus melanarius*, *Harpalus amputatus*, and *Amara littoralis*. Sensory manipulations and multiple-choice seed feeding bioassays showed olfactory perception of seed volatiles as the primary mechanism used by omnivorous carabids to detect and distinguish among seeds of *Brassica napus*, *Sinapis arvensis*, and *Thlaspi arvense* (Brassicaceae). Seed preferences differed among carabid species tested, but the choice of desirable seed species was generally guided by the olfactory perception of long chain hydrocarbons derived from the seed coat surface. These olfactory seed cues were essential for seed detection and discrimination processes to unfold. Disabling the olfactory appendages (antennae and palps) of carabid beetles by ablation left them unable to make accurate seed choices compared to intact beetles.

[1] Plant Sciences Department, College of Agriculture and Bioresources, University of Saskatchewan, Saskatoon, Saskatchewan, Canada. [2] Department of Agricultural, Food and Nutritional Science, University of Alberta, Edmonton, Alberta, Canada. ✉email: kaa316@mail.usask.ca

Biological control (biocontrol) is an important service provided by insects in both natural and managed ecosystems[1]. In many cases, biocontrol agents (i.e., natural enemies), including insect predators and parasitoids, are endemic in agricultural fields and thus offer natural control services[2]. In agroecosystems, natural control services help maintain ecosystem balance and reduce reliance on pesticide inputs, which in turn helps to mitigate pesticide resistance problems[3]. To determine and promote the beneficial services that predators provide in the ecosystem, it is essential to understand the basic interactions between insect predators and the organisms they target in the field[4,5]. One of the first steps is to determine how insect predators perceive their target organisms and assess their suitability for feeding. Such knowledge could be used to better predict the efficacy of predators as natural biocontrol agents under field conditions[4,6]. This information becomes even more crucial when the predatory species of interest are omnivorous and able to feed on a wide array of foods[4,7]. Therefore, it is vital to elucidate the sensory mechanisms of prey detection and discrimination to better understand what renders certain species more prone to elevated predation risks, especially when other species that serve as alternative food sources are accessible to the predator[8]. Understanding the core ecological aspects of feeding habits in predatory insects will allow agroecosystems to be managed for improved diversity and abundance of insect predators, thereby enhancing their ecological functioning[2].

Ground beetles (Coleoptera: Carabidae) are one of the most important groups of predatory insects in temperate ecosystems and function as epigaeic polyphagous predators[9]. Carabid activity can be diurnal or nocturnal depending on species identity and/or habitat properties[10]. They are generally voracious feeders able to consume close to their body weight of food each day[11]. Numerous carabid species prey upon various agricultural pests such as aphids[12], lepidopteran caterpillars[13], dipteran eggs and midges[14], wireworms[15], and slugs[16]. In addition to pests, numerous species of predatory carabids also feed on seeds of different weed species after seed shed[17]; thus, the majority of carabid predators are omnivorous. These diverse feeding habits among carabids make them amongst the most formidable predators in the agroecosystem, where they have been reported to regulate populations of various pests and weeds[18,19].

Despite the high potential of omnivorous carabids to provide biocontrol in agroecosystems, their complex feeding habits make it difficult to predict their biocontrol efficiency. Seed feeding in particular remains poorly understood, and it is unclear why seed feeding habits evolved given the abundance of prey in arable fields[20,21]. Therefore, it is difficult to predict which species of seeds or prey would be prone to elevated carabid attacks under realistic situations. Two hypotheses have been proposed to explain seed feeding habits in omnivorous carabids: (1) omnivorous carabids seek consumption of seeds only when prey species are scarce or inaccessible; or (2) omnivorous carabids mix both food types because prey feeding alone is insufficient for survival and development[8,22]. It was first thought that seed feeding would be biologically important to only to a small group of granivorous carabids species that subsist on a diet composed mainly of specific seeds[23,24]. Beyond that, seed feeding habits in omnivorous carabids would be opportunistic and tend to arise mostly when alternative foods are scarce or hard to obtain[8]. Seed feeding habits in carabids often transcend the artificial limits imposed by the dietary specialization reasoning (omnivory vs. granivory), as seeds are featured in the diets of a large number of carabid taxa[25,26]. This evidence suggests that seed feeding habits in carabid species have evolved due to yet unexplored biological needs not necessarily exclusive to granivorous carabids *sensu stricto*. Therefore, it is vital to explore the ecology of seed feeding habits in omnivorous carabids and elucidate their impact on the ecological functioning of carabids in agroecosystems.

Seed feeding in omnivorous carabids is likely driven by behaviors that are directed towards addressing specific biological needs. Random seed encounters were assumed to be the sole driver of seed feeding and seed selection decisions in carabids[8]. However, field and laboratory studies have demonstrated that carabid predators show active selection of specific seed species when multiple seed species are presented[27–29]. Such preferences require carabids to discriminate among seeds of different species, assess their suitability aspects, and then choose the most desired species[30]. To perform these tasks, carabids need to collect reliable, seed-derived information through their different sensory systems[31]. Our current understanding of the sensory and behavioral mechanisms that underlie the discrimination of, and preference for seeds in carabids remains rudimentary[32]. Seed odors can influence carabid orientation responses in olfactometers[33,34], but it remains uncertain whether chemoperception alone can drive accurate decision-making in seed selection. It is plausible that the choice of suitable seed species cannot take place without sensory inputs from the visual and/or gustatory systems. This uncertainty makes it difficult to ascertain the sensory cues that enable carabid predators to accurately identify suitable seed species. It is thus essential to study the sensory ecology of seed detection and discrimination in omnivorous carabids to better understand which seed properties (chemical or physical) may render seeds of specific species more vulnerable to carabid attacks[35]. This is expected to further our knowledge around the ecological functioning of carabid predators in agroecosystems, and potentially also help us identify the biological and ecological factors behind the evolution of seed feeding habits in omnivorous carabids[36].

The dearth of detailed sensory studies coupled with the complexity of feeding habits in carabid predators make it difficult to decipher the sensory ecology of seed detection and discrimination[10,24]. Prey hunting behaviors in carabids are often driven by visual cues, yet these behaviors tend to break down if prey items are immobilized[37]. Carabid visual receptors thus seem more attuned for detecting prey movement and should be more helpful for hunting down highly mobile prey. Carabids are, therefore, not expected to rely on visual receptors to detect sessile prey or weed seeds, especially that seeds are usually cryptic and difficult to distinguish against the soil[36]. Alternatively, carabids are expected to rely on their chemoreceptors to detect and distinguish among different species of seeds and/or immobile prey. Indeed, mechanistic studies have shown that chemoperception is the primary sensory mechanism that underlie prey detection and selection in both carnivorous (specialized) and omnivorous (unspecialized) carabid predators[38–40]. Chemoreceptors located on the antennae and palps of larval and adult carabids have been found to detect prey-derived volatile chemicals (i.e., prey odors), and guide the selection of suitable or desirable prey species[41–43]. Similar mechanistic knowledge about seed detection and discrimination is still wanting in the carabid literature. Thus, it remains to be determined if the perception of seed odors is also the primary sensory mechanism behind seed preferences in carabid seed predators.

Here, we explore the sensory ecology involved in seed perception and recognition in carabid seed predators. The main objective was to elucidate the sensory mechanisms of seed perception, and identify the sensory cues that carabids exploit to detect and distinguish among seed different species. To that end, we used *Poecilus corvus* (Leconte), *Pterostichus melanarius* (Illiger), *Harpalus amputatus* Say, and *Amara littoralis* Dejean (Coleoptera: Carabidae) omnivorous carabid species, and seeds of *Brassica napus* L., *Sinapis arvensis* L., and *Thlaspi arvense* L.

(Brassicaceae) as a model system. Omnivorous feeding in the current study is used to refer to carabid species that are able to feed on both plant and animal foods, irrespective of their dietary breadth or feeding specialization[4,24]. Sensory manipulation protocols coupled with multiple-choice seed feeding bioassays identified olfactory perception as the primary sensory mechanism used by carabids to detect and discriminate among seeds of different species. The choice of suitable seed species was driven by the perception of long-chain hydrocarbons (volatile chemicals) derived from the epicuticular lipids located on the seed coat surface. The seed surface hydrocarbons isolated and identified here seem to encode chemical information about the fatty acid composition of seed species. Carabid predators are thus predicted to exploit the information encoded in seed surface hydrocarbons to assess the fatty acid composition of the seed species and then, choose the suitable seed species based on desirable lipid content[20,21]. Based on this, lipid-rich seed species are likely more prone to elevated carabid predation in the field, given that seed-handling costs, as determined by seed physical properties[28,29], are not widely variable among the seed species accessible to carabids.

## Results

**Olfactory chemoreceptors enable seed detection and discrimination in carabid seed predators.** Seeds of three brassicaceous species were offered to three species of sensory-manipulated carabid predators. Total seed consumption was used as a measurement of carabid seed detection success under sensory manipulation treatments. Analysis of variance (ANOVA) showed that disabling different sensory organs significantly affected the ability of carabid species to detect seeds in the feeding arenas (ANOVA: $F_{16,530} = 1.945$, $P = 0.015$; $n = 65$, $\alpha = 0.05$; Fig. 1a–c). There were no significant differences in the responses of carabids based on sex (ANOVA: $F_{1,530} = 0.024$, $P = 0.87$, $n = 65$, $\alpha = 0.05$). Within species, covering the compound eyes of *P. corvus* caused a significant reduction (ca. 30%) in seed detection success compared to intact insects of the same species (ANOVA: $F_{8,207} = 17.90$, $P < 2e{-}16$, $n = 25$, $\alpha = 0.05$; Fig. 1a). Carabids having functional eyes only failed to find seeds, and their seed consumption was not significantly different from zero. Carabids with antennae and/or maxillary palps (olfactory organs) left intact found seeds with considerable success, displaying 30–50% decrease in consumption rates compared to intact beetles. Intriguingly, when predators were left with only labial palps fully intact (gustatory organs), seed detection success was not statistically different from zero. Similar response patterns were observed for both *P. melanarius* (ANOVA: $F_{8,206} = 19.244$, $P < 2e{-}16$, $n = 25$, $\alpha = 0.05$; Fig. 1b), and *A. littoralis* (ANOVA: $F_{8,117} = 16.41$, $P = 4.36e{-}16$, $n = 15$, $\alpha = 0.05$; Fig. 1c). Fully intact antennae and palps in both species accounted for 50–75% of seed detection success, while those with intact eyes only were not successful.

Comparing seed choice responses with mixed models revealed significant impacts of sensory manipulations on seed selection in the three carabid species under study (Mixed Effects: $F_{16,1533} = 2.267$, $P = 0.00287$ $n = 65$, $\alpha = 0.05$; Fig. 1d–f). Responses did not differ between males and females for any of the carabid species tested (Mixed Effects: $F_{1,1548} = 0.0013$, $P = 0.97151$, $n = 65$, $\alpha = 0.05$). There was a significant interaction between sensory treatments and carabid species (Mixed Effects: $F_{16,1552} = 2.192$, $P = 0.004173$, $n = 65$, $\alpha = 0.05$). *Poecilus corvus* showed a strong preference for *B. napus* seeds, and this preference was maintained across all treatments where beetles had two intact antennae and/or four intact palps (Mixed Effects: $F_{8,621} = 15.169$, $P < 2e{-}16$, $n = 25$, $\alpha = 0.05$; Fig. 1d). When antennae were ablated, beetles needed all four palps intact to

make an accurate seed choice. The ability for choosing desirable seeds was lost when beetles were left with only one pair of either palps; preference for *B. napus* seeds lost statistical significance under this treatment. Accurate seed choice was also lost when antennae and palps were ablated, leaving the beetles with functional eyes only. Similar response patterns were observed for both *P. melanarius* (Mixed Effects: $F_{8,601} = 26.449$, $P < 2e{-}16$, $n = 25$, $\alpha = 0.05$; Fig. 1e), and *A. littoralis* (Mixed Effects: $F_{8,336} = 15.88$, $P < 2.2e{-}16$, $n = 15$, $\alpha = 0.05$; Fig. 1f). Antennae and/or both types of palps left intact enabled beetles of both species to make an accurate seed choice compared to the positive control, but the two species showed clear differences in their seed preference. *Pterostichus melanarius* had a strong preference for *B. napus* seeds, whereas *A. littoralis* preferred seeds of *T. arvense*.

**The species-specific chemical cues necessary for seed discrimination are located on the seed coat surface.** The previous experiment established that carabid seed predators rely on their chemoreceptors to detect seeds of different species, and also to choose the most preferable seed species among them. We originally attempted sampling the headspace of the three brassicaceous seed species via solid-phase microextraction (SPME) fibers or dynamic air entrainment using Porpak Q, but this failed to detect the seed volatiles necessary for seed discrimination in carabids. Direct extraction of seed surface chemicals yielded the candidate species-specific seed volatile chemicals necessary for seed discrimination (Fig. 2a–c). Seed volatiles were composed of fatty acid derivatives comprising three main groups of long-chain aliphatic lipids: alkanes, esters, ketones. Seed species showed significant differences in their profiles of volatile chemicals (Mixed Effects: $F_{10,72} = 7.008$, $P = 1.428e{-}07$, $n = 5$, $\alpha = 0.05$). *Brassica napus* seeds featured the simplest profile of surface chemistry, with only two major compounds in their profile (Fig. 2a, see also Table 1). By contrast, surface chemistries of *S. arvensis* and *T. arvense* seeds showed more complex profiles of alkanes, ketones, and esters (Fig. 2b, c, see also Table 1). Fatty acid ethyl esters were not commercially available and hence further research is needed to confirm their structure.

**Seed surface chemistry can drive seed selection decisions in carabid seed predators.** The seed surface volatile chemicals (i.e., hydrocarbons) isolated and identified in the previous experiments were used to coat the surface of specific seed species to test if chemical coating could alter the palatability of seeds to carabid predators. In the first set of experiments, seeds of *B. napus* were coated with surface extracts of *T. arvense* seeds, as these two seed species feature widely different profiles (simple vs. complex) of hydrocarbons (see above). Seeds of *B. napus* became significantly less preferable to *P. corvus*, *H. amputatus*, and *A. littoralis* when coated with *T. arvense* surface extracts (Mixed Effects: $F_{2,48} = 5.284$, $P = 0.00842$, $n = 30$, $\alpha = 0.05$; Fig. 3a). In the second set of experiments, seeds of *T. arvense* seeds were coated with surface extracts of *B. napus*. Coated *T. arvense* seeds were considerably more acceptable to *P. corvus*, *H. amputatus*, and *A. littoralis*, although this change in preference was not statistically significant (Mixed Effects: $F_{2,48} = 0.0427$, $P = 0.8372$, $n = 30$, $\alpha = 0.05$; Fig. 3b). These results suggest that carabids seem to choose the seed species with the simplest surface chemistry when determining the most preferable seed species, assuming seed size is similar among species.

## Discussion

Our study has demonstrated that omnivorous carabid seed predators rely mainly on chemoperception to detect and choose among different seed species. The sensory information needed for

**Fig. 1 Identification of the carabid sensory systems involved in seed detection and discrimination processes.** Seed detection success of sensory-manipulated *Poecilus corvus* (**a**), *Pterostichus melanarius* (**b**), and *Amara littoralis* (**c**) beetles as measured by total number of seeds consumed (mean total seed consumption ± mean standard error). Seed choice responses of sensory-manipulated *Poecilus corvus* (**d**), *Pterostichus melanrius* (**e**), and *Amara littoralis* (**f**) beetles as measured by numbers of seeds consumed from each species (mean number of seeds consumed ± mean standard error). (+/+): positive control (intact insects); (+/−): unilateral control (half sensory capability); (−/−): negative control (zero sensory capability).

these tasks was shown to be encoded in hydrocarbons (volatile chemicals) located on the seed surface and is detected by the chemoreceptors located on the antennae and palps. Chemoperception has been reported to guide essential aspects of prey searching and detection in omnivorous and carnivorous carabids[38,42]. Visual cues did not elicit the sensory response necessary to guide seed choice. Therefore, visual seed characteristics such as seed color and surface texture do not seem to play influential roles in shaping carabid seed selection decisions, or at least in the species tested. This is logical as seeds are sessile and usually scattered on the soil surface, or even buried underneath it[44], which presumably renders visual detection of seeds by carabid predators difficult. Similarly, visual perception can be unreliable for detection of sessile prey in carnivorous

carabids[37,45]. Carabid visual receptors may be more tuned towards detecting prey movement, and are expected to be more useful for hunting down mobile prey than locating sessile insect prey or seeds[46]. It is important to mention here that sensory manipulations, although intrusive, did not appear to cause considerable impairment to the seed selection ability of the carabids under study. Sensory-manipulated carabids carrying sufficient functional, intact chemoreceptors (antennae, palps, or both) were still able to exhibit accurate seed choices akin to those of fully intact insects (positive control). Sensory manipulations as such did not seem to affect the ability of carabids for information processing or decision-making. Insect sensory appendages carry receptors that collect information from the surrounding environment, but do not play a major role in processing the sensory

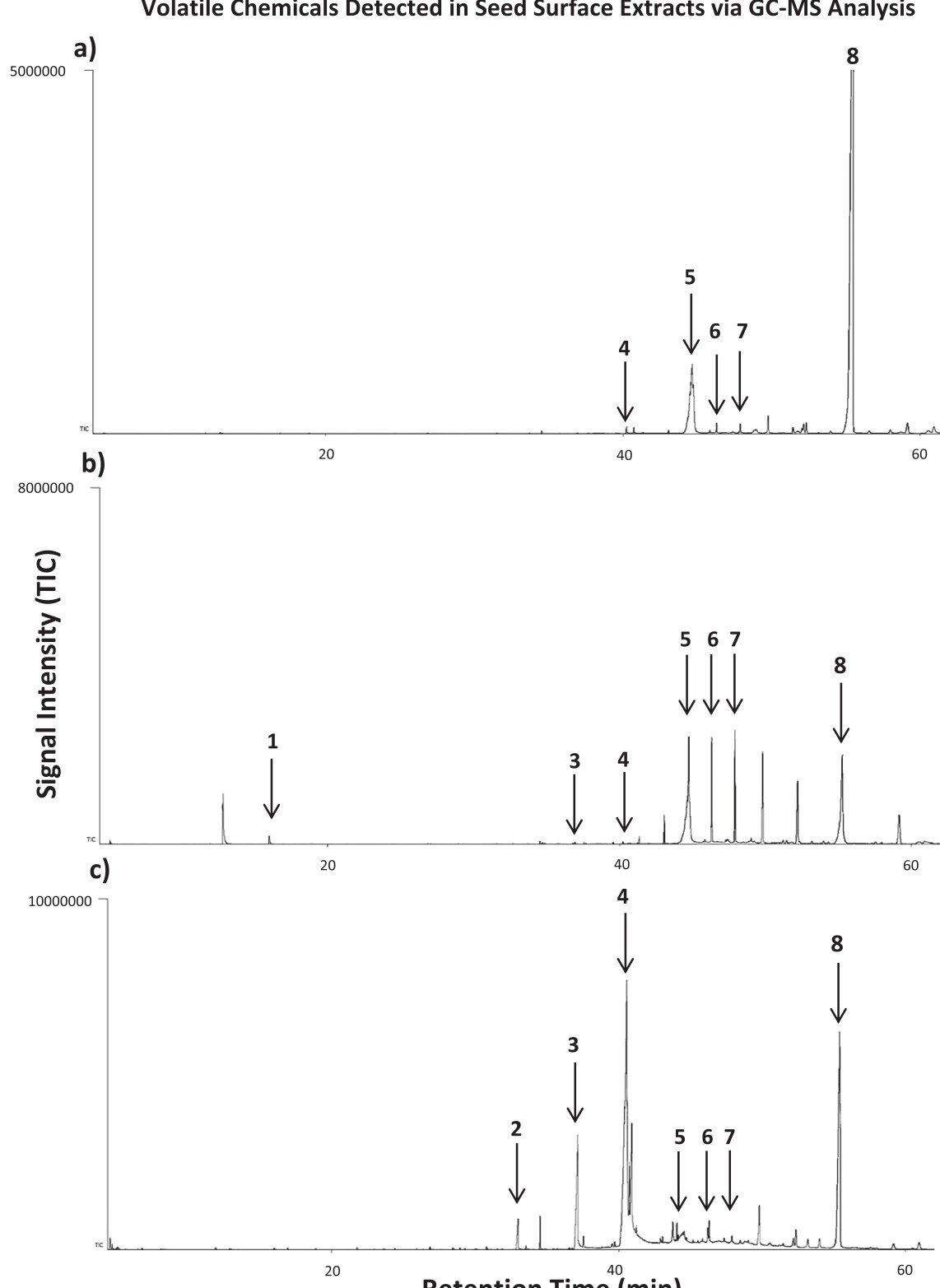

**Fig. 2 Volatile chemical compounds detected in surface extracts of the three brassicaceous weed seed species.** Seed volatile chemical compounds detected in the surface seed extracts of *Brassica napus* (**a**), *Sinapis arvensis* (**b**), and *Thlaspi arvense* (**c**) measured as total ion content TIC (ion abundance) in mV. Numbers represent compounds: (1) Nonanal, (2) n-Tetradecanoic acid, (3) Hexadecanoic acid ethyl ester, (4) E-9-Octadecanoic acid ethyl ester, (5) Hexacosane, (6) Hepatacosane, (7) Nonacosane, (8) 15-Nonacosanone.

**Table 1 Seed surface volatile chemicals isolated from the three brassicaceous species used in the experiments showing identity and average percentage of the detected compounds on measurements of peak areas in GC-MS chromatograms.**

| Chemical compound | | | | Weed species | | | |
|---|---|---|---|---|---|---|---|
| | RT | Formula | CAS # | Brassica napus (n = 5) | Sinapis arvensis (n = 5) | Thlaspi arvense (n = 5) | |
| Nonanal | 15.97 | $C_9H_{18}O$ | 124-19-6 | ND | 1.97 ± 0.3% | ND | |
| n-Tetradecanoic acid | 33 | $C_{14}H_{28}O_2$ | 544-63-8 | ND | ND | 2.8 ± 0.5% | |
| Hexadecanoic acid ethyl ester | 37.23 | $C_{18}H_{36}O_2$ | 626-97-7 | ND | <1% | 4.63 ± 1.88% | * |
| E-9-Octadecanoic acid ethyl ester | 40.73 | $C_{20}H_{38}O_2$ | 6114-18-7 | <1% | <1% | 43.3 ± 7.76% | * |
| Hexacosane | 44.64 | $C_{26}H_{54}$ | 630-01-3 | 18 ± 1.45% | 11.42 ± 2.87% | <1% | * |
| Heptacosane | 44.87 | $C_{27}H_{56}$ | 593-49-7 | <1% | 31.47 ± 4.44% | <1% | * |
| Nonacosane | 47.43 | $C_{29}H_{60}$ | 630-03-5 | <1% | 17.03 ± 2.17% | <1% | * |
| 15-Nonacosanone | 55.42 | $C_{29}H_{58}O$ | 2764-73-0 | 79.98 ± 1.73% | 32.46 ± 6.29% | 45.66 ± 5.3% | * |

RT retention time in minutes, CAS # Chemical Abstracts Service Registry Number in NIST Mass Spectral Library, ND not detected.
*Indicates significant quantitative differences between volatiles of weed seed species as measured by peak area.

**Fig. 3 Impact of seed surface hydrocarbons on carabid seed preference.** Feeding responses (mean number of seeds consumed ± mean standard error) of three carabid species offered seeds of B. napus coated with the surface chemicals of T. arvense against uncoated intact B. napus seeds (**a**), and seeds of T. arvense coated with the surface chemicals of B. napus seeds against uncoated T. arvense (**b**) in two-choice feeding bioassays.

input or inducing behavioral responses[47,48]. Higher cognitive centers like optic lobes, antennal lobes, lateral horns, and mushroom bodies are responsible for processing the sensory input and inducing appropriate behavioral responses[49,50], and those remained intact in the carabids under study. Gender of the carabid species (i.e., males vs. females) also did not seem to affect their responses to sensory manipulations throughout the study. Gender of carabid predators usually has no considerable impact on sensillar packages[51]. We therefore expect that seed perception mechanisms in carabid seed predators are unlikely to be sexually dimorphic.

The carabid species we studied carry most of the chemoreceptors responsible for seed detection on their antennae[51–53]. Antennae, either alone or in combination with palps, enabled carabids in our study to identify the suitable seed species with high accuracy. By contrast, predators carrying one functional, intact pair of either set of palps failed to make accurate seed choices. Carabid antennae usually carry an abundance of olfactory receptors that enable predators to collect specific chemical information about their food[54,55]. Olfactory receptors can also be found on maxillary and labial palps of insects, but their abundance on these appendages is usually low[56]. On the contrary, gustatory receptors are often more abundant on maxillary and labial palps[57,58]. Olfactory and gustatory receptors show considerable similarities in their structures and physiological

functioning, and collect chemical information of similar nature[59]. Nonetheless, the sensory information perceived through olfactory receptors is frequently suggested to be more accurate and specific than information perceived via gustatory receptors[60]. These lines of reasoning might explain why accurate seed choice took place in all treatments where antennae were not ablated from carabid heads. It might also explain why all four maxillary and labial palps were needed for accurate seed choices. Beetles carrying either type of palps alone did not seem to perceive the chemical information necessary for accurate seed choices. Given our findings we posit that olfactory receptors are, in all likelihood, the type of chemosensilla responsible for seed perception in carabid seed predators.

We have shown that the chemical cues that enable carabid predators to select between seeds of different species are composed of fatty acid derivatives. The fatty-acid-derived seed cues isolated and identified in this study are located on seed surfaces and comprise three main groups of long-chain aliphatic lipids: alkanes, esters, ketones. The three seed species tested showed qualitative and quantitative differences in their profiles of seed surface chemicals. But the quantitative differences in alkanes, ketones, and esters accounted for the majority of species-specific differences among seed species. Simple profiles of surface volatiles like B. napus were more preferable than complex profiles, based on seed coating experiments; complex surface chemistry (e.g., T.

*arvense* seeds) seems to encode information that deters feeding. No glucosinolate compounds were detectable in extracts of seed surface chemicals, even though seeds of brassicaceous species usually harbor considerable amounts of these defensive compounds[61]. Other authors have also reported that glucosinolates and their breakdown products (isocyanates) are not usually detectable in headspaces of brassicaceous species, or in extracts of their epicuticular lipids[62,63]. These findings contradict the belief that glucosinolates may act as deterrents against carabid seed predation[64]. *Amara littoralis* in our feeding multiple-choice experiments showed a strong preference for *T. arvense* over *B. napus*. This strong preference for *T. arvense* seeds was probably not based on seed defensive chemicals (see below), as seeds of *T. arvense* usually contain high levels of glucosinolates[65]. We propose that glucosinolates and their breakdown products are unlikely to function as preingestive seed feeding deterrents for carabid seed predators. Instead, seed surface hydrocarbons are more likely to act as the main preingestive signaling chemicals that carabid predators exploit to guide their seed foraging behaviors. It remains uncertain if the same applies to non-brassicaceous seeds.

Hydrocarbons are an essential constituent of the cuticle layer that covers surfaces of both somatic and reproductive plant tissues including seeds, and consequently serve wide ecological functions[66,67]. Behavioral testing of the identified seed surface chemicals via coating seed species showed that seed surface hydrocarbons were able to drive feeding responses of carabids. It is possible the presence of fatty esters among the surface hydrocarbons of *S. arvensis* and *T. arvense* seed made these species less palatable to the carabid species tested. Further research is still needed to validate this assumption, but it is plausible since coating *B. napus* seeds with surface extracts of *T. arvense* seeds rendered *B. napus* seeds remarkably less preferable to all carabid species tested. Seeds of *B. napus* lack fatty esters, but the coating procedure introduced them onto *B. napus* seeds, rendering them less palatable. Coating *T. arvesne* seeds with surface extracts of *B. napus* seeds did not greatly change their palatability to carabids seeds, likely because fatty esters were already present. The feeding stimulatory effects observed for seed surface chemicals in the behavioral experiments with chemically-coated seed species validates our previous conclusion and fits into an ample body of evidence documenting plant surface hydrocarbons as interlocutors of feeding and oviposition preferences in insects[68]. For instance, surface lipids have been shown to mediate essential aspects of feeding behaviors and host plant choice in insect herbivores of Diptera[69], Lepidoptera[70], Coleoptera[71], Thysanoptera[72], Hymenoptera[73], and Hemiptera[74]. Plant surface hydrocarbons, therefore, provide the "kairomonal" signals necessary for the ecological interactions between insect and plant species to initiate and unfold[64]. By the same token, carabid predators seem to exploit seed surface hydrocarbons as the kairomones necessary for guiding their seed foraging behaviors.

The kind of information carabid seed predators extract from seed kairomones remains unknown, but there are well-documented—yet poorly understood—correlations between plant species identity, cellular fatty acid metabolic-biosynthetic pathways, and composition of seed surface hydrocarbons[66,75]. It is highly possible that seed surface hydrocarbons carry information about the fatty acid composition of the seed, and potentially also of their quantity or quality. Carabid predators seem to exploit the information encoded in seed surface hydrocarbons to assess the fatty acid composition of the seed species and identify the seed species with desirable lipid content[20,21]. Microbial volatile emissions that might have originated from the seed microbiome were minimal in the current study, and were thus unlikely to have impacted the results. Microbial volatiles are often of less than $C_{15}$ and of low molecular mass (33 Da)[76], which was not the case for

the majority of seed volatiles isolated and identified in this study. The seed chemicals identified here are derived from long-chain fatty acids ($C_{18}$-$C_{29}$). Signaling compounds of such high molecular weight are generally low in volatility and can act at close ranges only[68]. This likely explains why carabid species included in this study were observed to 'touch' or 'contact' the seed surface with their antenna and palps prior to seed selection[77]. It might also explain why the seed headspace was devoid of any species-specific volatiles in static and dynamic headspace sampling experiments in our previous work[77].

The information encoded in seed surface hydrocarbons appears vital for carabid interactions with seeds as such interactions could not take place when carabid predators were stripped of their olfactory chemosensilla. Based on this, we propose that carabid beetles employ olfactory templates or search images to guide their seed feeding behaviors. It remains to be explored if the formation of olfactory templates that guide seed recognition and choice may be influenced by learning and experience, be it non-associative or associative learning mechanisms[78]. Olfactory priming of carabid seed predators with odors of specific seed species had no effect on seed choice responses in carabids as a mechanism of non-associative learning[77]. Seed odors alone are, therefore, not expected to be the sole driver of seed selection decisions in omnivorous carabid predators. Alternatively, seed preferences in omnivorous carabids could be guided by olfactory templates or search images created by associating seed odors with seed handling parameters. Therefore, the active selection of nutritious seed species via seed odor (see above) is unlikely to explain seed preferences in cases where seed handling parameters may differ among seed species[30]. Further research is needed to validate this hypothesis, and explore the roles of learning and experience in the ecology of seed feeding habits in carabids.

It should be noted here that seed selection decisions were studied under controlled conditions in the laboratory. Seed selection decisions by the carabids tested under these artificial experimental conditions are likely to differ from the more complex, realistic conditions[79]. Furthermore, the chemical coating procedures in the behavioral experiments were carried out on seeds of the same seed species. Thus, the effects of seed characteristics other than seed surface chemistry were minimal in the seed coating study. Therefore, data in the current study cannot accurately predict seed preferences when seed species vary not only in their surface chemistries, but also in other properties such as size, mass, and coat hardness. This does not mean that carabids will not exploit olfactory seed cues to guide their seed selection decisions under realistic situations. Rather, carabids will still rely on olfactory seed cues to recognize the different seed species in the field, but the selection of suitable seed species may not always be driven by the nutritional quality of the seed per se. The active selection of desirable seeds can be constrained, or even abandoned altogether, depending on habitat properties and/or composition of the seed bank[80–82]. The biotic and abiotic properties of carabid habitats affect the abundance of plant and animal foods, as well as the microclimatic and microsite conditions[83,84], which can profoundly affect the composition and structure of the carabid community in the field. This, in addition to factors relating to physical seed traits, fear, and dominant species in the carabid community can all render the search for desirable seed species more laborious or less rewarding to carabids foraging under field conditions[85,86]. Carabids in such case are more likely to choose seed species that are more accessible or easier to handle[30], irrespective of their nutritional quality. *Amara littoralis* in the current study showed a strong preference for *T. arvense* seeds, which was an observation potentially driven by size relationships since both *A. littoralis* and *T. arvense* were the smallest carabid and seed species in the experiment, respectively[87]. Despite

this, the ability of *A. littoralis* to accurately choose seeds of *T. arvense* was totally lost when this carabid species was stripped of its olfactory appendages (antennae and palps ablated). Thus, olfactory seed cues appear to be an essential requirement for the seed detection and discrimination processes to unfold, even though the ultimate choice of preferable seeds may not always be driven by the nutritional quality of seed species.

All carabid species included in this study are usually described as omnivores, feeding on both plant and animal foods[24]. However, these species often show differences in their dietary specializations, with *H. amputatus* and *A. littoralis* being more specialized toward seed feeding than *P. melanarius* and *P. corvus*[4,24]. Such differences in dietary specialization may affect which seed species carabids choose more preferably for consumption in choice tests[29], but exploring this was not an objective in the current study, and further research is needed to elucidate this. Still, the differences in dietary specializations did not seem to entail any considerable divergences in the sensory mechanisms that underlie seed recognition in the carabid species tested. All carabids tested employed olfactory perception of seed surface hydrocarbons as the primary sensory mechanism of seed detection and discrimination. Overall, our findings prove that seed selection decisions in carabids are guided by seed-derived chemical cues that are detected by chemoreceptors located on the antennae and palps of carabid predators, although the nature of this relationship may be sensitive to multiple biotic and abiotic factors in the local environments.

## Methods

**Seed material.** Seeds of three different brassicaceous species (Brassicaceae: *Brassica napus*, *Sinapis arvensis*, *Thlaspi arvense*) were used as model species in this study. Seeds of these three species are similar in color and surface texture. Previous studies have shown that these seed species varied widely in their palatability to carabid seed predators. Canola seeds (*B. napus*) in those studies were highly preferable to carabid seed predators (~65% seed removal rate per week), whereas seeds of wild mustard (*S. arvensis*) and field pennycress (*T. arvense*) were moderately (~55% seed removal rate per week) and weakly preferable (~25% seed removal rate per week), respectively[27,32]. Accordingly, seeds of *B. napus* were used as a highly preferable seed species in this study, whereas seeds of *S. arvensis T. arvense* represented moderately and weakly acceptable seed species. Seed masses of the three weed species were obtained from stored samples collected in summers of 2016–17. Seeds were collected from different field sites at the Kernen Crop Research Farm near Saskatoon, SK, Canada (52°09'10.3″ N 106°32'41.5″ W), and then stored at 5 °C to be used the next year after collection. Seeds were not dried before storage, and contact between the skin and seed surfaces was avoided by wearing gloves and using sterile tweezers during seed handling.

**Carabids.** Adults of the omnivorous carabid species *Poecilus corvus*, *Harpalus amputatus*, *Pterostichus melanarius*, and *Amara littoralis* were used in this study as all are known to consume weed seeds. Live adults were collected from different field sites at the Kernen Crop Research Farm in the summers of 2018–20 via dry pitfall trapping. Field sites chosen for carabid trapping were seeded with canola, legume, or cereal crops. Pitfall traps consisted of two plastic 0.5 L cups (10 cm height × 8 cm diameter). One cup acted as a sleeve and with its lip kept flush with the soil surface, and the other cup (the actual trap) was inserted into it[88]. Pitfall traps were enclosed into cages of fine wire mesh ($\sigma = 1.1$ cm) to prevent vertebrates from entering the traps and ravaging the catches. Traps were emptied every three days and the collected insects were placed into plastic boxes (40 cm × 25 cm, 25 cm depth) lined with plant material and moist filter paper. Boxes were transported to the laboratory for identification and experimentation. Carabid species identity and sex of the experimental carabids were determined using keys in Lindroth (1961–1969)[89].

The abundance of *P. melanarous*, *P. corvus*, *H. amputatus*, and *A. littoralis* in the catches fluctuated among the seasons. Therefore, the carabids species used in the behavioral experiments were not the exact same species used in the sensory experiments. Still, all of the carabid species used in this study feature omnivorous feeding habits, irrespective of any differences in their dietary specialization[24]. Thus, the variability of species in some of the experiments is unlikely to undermine the validity of our ecological inferences given the omnivorous nature of the included carabid species[4,24]. The impact of dietary breadth on seed preference was not investigated in this study.

**Elucidating the sensory mechanisms of seed detection and discrimination in carabids.** Seeds of the three brassicaceous weed species were offered to carabid

species in multiple-choice feeding bioassays. Feeding arenas were made from a large Petri dish (Ø = 25 cm, 5 cm depth) lined with a 2-cm layer of sterilized, moist sand as a neutral and easy-to-sterilize substrate. Seeds were placed into plastic tray rings (Ø = 28 mm, 6 mm depth) filled with white plasticine and positioned near the perimeter of the Petri dish. Plasticine has been shown not to interfere with seed preference in carabid seed predators[87]. The same type of white plasticine (Sargent, Thailand) was used across the experiments to avoid any confounding effects that might arise from using different types of plasticine. A total of three trays each harboring 25 seeds of one species were placed into each Petri dish so that the seed patch was level with the sand layer. Imbibed seeds were used for all the feeding experiments. Seeds were imbibed by placing seeds on wet filter paper in Petri dishes (Ø = 6 cm, 2 cm depth), and leaving seeds to absorb moisture for 24 h in a growth chamber at 21 ± 1 °C[27].

Beetles were not fed after collection and were starved for 72 h prior to feeding experiments to empty their guts and standardize their hunger level[34]. Beetle starvation was carried out by placing a single beetle (to prevent cannibalism) into a clean and sterile Petri dish (Ø = 6 cm, 2 cm depth) lined with a moist filter paper. Petri dishes were then covered and incubated in a growth chamber at 21 ± 1 °C and 16:8 L:D photoperiod[90]. The 72-h period was also sufficient for any olfactory memory that might have formed while beetles had been foraging in the field to decay[91]. After 72 h of starvation, the Petri dishes were placed in a refrigerator at 5 °C for 20 min to reduce the activity of the carabid predators prior to sensory treatment.

Sensory manipulation was conducted by placing the immobilized beetle in a dissection plate, and then ablating one or more of the sensory appendages (i.e., antennae, maxillary palps, and/or labial palps) under a stereoscope. Ablation of sensory appendages is widely used for sensory and behavioral studies in insects[92,93]. In this study, ablating antennae of carabid beetles enabled the creation of treatment groups that lacked the ability to smell (no olfaction capability). On the other hand, ablating the maxillary and/or labial palps produced treatment groups that lacked the ability to taste (no gustation ability). In treatment groups where vision needed to be disabled, carabid beetles were blinded by covering their compound eyes with permanent black ink (no visual capability)[94]. Sensory treatments as above (i.e., ablation and blinding) enabled the creation of different groups of carabid beetles each lacking the ability to perceive sensory information of a specific nature; visual, olfactory, or gustatory (see Table 2).

Seed feeding responses of the sensory-manipulated carabids were then compared to three control groups: positive, unilateral, and negative. The positive control represented 'fully intact' beetles with a full complement of functional sensory organs. Beetles in the unilateral control had their antenna, maxillary palp, and labial palp ablated and the compound eye covered only on one side of the head. The side on which the sensory treatment was carried out (left or right) was randomized to avoid bias. Finally, the negative control group was created by ablating all of the sensory organs on the carabid head and blackening both compound eyes, creating beetles with no visual or sensory capabilities.

Sensory-manipulated carabids were given 10 min to 'acclimatize' and were then a single beetle was released into each feeding arena (Petri dish). Feeding arenas were covered and incubated in a growth chamber at 21 ± 1 °C, and 16:8 L:D photoperiod and beetles were left to feed for 5 consecutive days[83]. At the end of the experiment, beetles were removed and the number of seeds consumed from each seed patch was recorded. Each beetle was used only once and all treatments and controls were replicated 25 times for both *P. corvus* and *P. melanarius*, and 15 times for *A. littoralis*. The sex ratio was close to 50♂:50♀ across all treatments and controls. Data were collected from feeding arenas in which beetles (treatments and controls) were alive at the end of the feeding time. Petri dishes (replicates) in which the beetles had died during the experimental time were discarded and repeated. As such, no data were collected from replicates with dead insects.

**Isolation and identification of the seed cues carabids exploit for seed recognition.** Results from the first study indicated seed choice differed among beetles lacking specific sensory organs, so we conducted a second study to identify and isolate seed cues mediating seed choice in carabids. Here, seed surface chemicals were extracted by placing 500 mg masses of imbibed seeds in clean and sterile 5 ml glass tubes. Following this, 3 ml of a 9:1 mixture of *n*-hexanes: dichloromethane (non-polar and polar solvents respectively) was added to each seed mass and shaken thoroughly for 15 min[95]. Preparations were then sealed with parafilm and incubated in a growth chamber at 21 ± 1 °C for 72 h. After incubation, the solvent mixture was removed and placed into a new sterile glass tube. All extracts were completely dried under a gentle stream of nitrogen, then re-eluted into 200 µl of *n*-hexanes[95] and stored at −80 °C until gas chromatography–mass spectrometry (GC-MS) analysis. Five independent extracts were carried out per seed species, and blank extracts (no seed added) served as negative controls.

Seed chemical extracts were analyzed by the GC-MS to identify any isolated volatile chemical compounds isolated. The GC-MS analysis was initiated by injecting aliquots of the volatile extracts (2 µl) into a HP-1 capillary GC column (50 m × 0.32 mm i.d., 0.55 µm film thickness) equipped with a cool on-column injector coupled to a mass spectrometer (JEOL AccuTOF 4 G, USA). The GC (Agilent 7890 A, USA) was programmed as follows: oven temperature was maintained at 50 °C for 2 min and then programmed at 5 °C/min to 250 °C, using helium as carrier gas. The initial identification of any detected compounds was

**Table 2 Treatment list for sensory treatments carried out on the carabid species under study with associated treatment descriptions.**

| Treatment number and code | Treatment description | |
|---|---|---|
| 1<br>Positive control (+/+) | Carabid beetles with<br>*fully functional sensory organs*<br>(intact carabid beetles) | Seeds of *Brassica napus* L.<br>Seeds of *Sinapis arvensis* L.<br>Seeds of *Thlaspi arvense* L. |
| 2<br>Unilateral control (+/−) | Carabid beetles with<br>*fully functional sensory organs on only one side of the body* | Seeds of *Brassica napus*<br>Seeds of *Sinapis arvensis*<br>Seeds of *Thlaspi arvense* |
| 3<br>Negative control (−/−) | Carabid beetles with<br>*all sensory organs blocked and removed*<br>(no sensory perception) | Seeds of *Brassica napus*<br>Seeds of *Sinapis arvensis*<br>Seeds of *Thlaspi arvense* |
| 4<br>(Antennae + Palps) | Carabid beetles with<br>*functional antennae and palps* | Seeds of *Brassica napus*<br>Seeds of *Sinapis arvensis*<br>Seeds of *Thlaspi arvense* |
| 5<br>(Antennae) | Carabid beetles with<br>*functional antennae* | Seeds of *Brassica napus*<br>Seeds of *Sinapis arvensis*<br>Seeds of *Thlaspi arvense* |
| 6<br>(Palps) | Carabid beetles with<br>*functional maxillary and labial palps* | Seeds of *Brassica napus*<br>Seeds of *Sinapis arvensis*<br>Seeds of *Thlaspi arvense* |
| 7<br>(Maxillary Palps) | Carabid beetles with<br>*functional maxillary palps* | Seeds of *Brassica napus*<br>Seeds of *Sinapis arvensis*<br>Seeds of *Thlaspi arvense* |
| 8<br>(Labial Palps) | Carabid beetles with<br>*functional labial palps* | Seeds of *Brassica napus*<br>Seeds of *Sinapis arvensis*<br>Seeds of *Thlaspi arvense* |
| 9<br>(Eyes) | Carabid beetles with<br>*functional compound eyes and ocelli* | Seeds of *Brassica napus*<br>Seeds of *Sinapis arvensis*<br>Seeds of *Thlaspi arvense* |

done by comparing retention indices (Kovats Index) of the detected peaks to published spectrum libraries. The TSS Utility software with a link to the NSIT Library was used for analyzing the chromatograms and identifying the detected peaks. Authentic samples of the compounds identified by GC-MS analysis were purchased from Sigma Aldrich (Sigma Aldrich, Canada) for structural confirmation.

**Testing the impact of seed surface hydrocarbons on seed choice in carabids**. The impact on seed foraging behaviors of seed cues identified above was tested via coating a specific seed species with surface extracts of a different seed species. The purpose of this experiment was to change the surface chemistry of a specific seed species via chemical coating and then, test if this would alter its palatability to carabid predators compared to uncoated (untreated) seeds of the same species. Treatments here represented coating seeds of *B. napus* with surface extracts of *T. arvense* seeds in one set of experiments, and coating seeds of *T. arvense* with surface extracts of *B. napus* seeds in another set of experiments. These two species showed wide differences in their profiles of surface hydrocarbons as shown by the previous experiment, which we expected was likely to induce measurable differences in seed selection responses of carabids.

Seed surface hydrocarbons were extracted with a 9:1 mixture of *n*-hexanes: dichloromethane as described in the previous section. The chemical coating was carried out by soaking the seeds in 2 ml of concentrated seed extract suspended in Triton X-100 (2% *v/v*) with for 30 min[96] then leaving the seeds for 15 min to dry at room temperature. Control uncoated seeds were soaked with 2 ml of *n*-hexanes (no seed extract) suspended in Triton X-100 (2% *v/v*) for 30 min then dried at room temperature for 15 min. Coated and uncoated seeds (25 seeds of each) were then placed in plasticine trays and offered in two-choice Petri-dish feeding bioassays to *P. corvus, H. amputatus,* or *A. littoralis* carabids following after 72 h of starvation. Petri dishes were kept in a growth chamber at 21 ± 1 °C and 16:8 L:D photoperiod. Feeding trials were replicated 10 times for each species, and insects were allowed to feed for 5 consecutive days. Carabids were removed at the end of the experiment and seed consumption rates were recorded.

**Statistics and reproducibility**. R v.4.0.3 (R Development Team 2020) was used for all data analysis. Three-way analysis of variance (ANOVA) was used to compare the total number of seeds consumed by each carabid over five days under the different sensory treatments. Significant differences between carabid species were detected when the full data set was analyzed, so data were analyzed and presented for each species separately. The analysis was carried out by fitting a maximal model including sensory manipulation, insect species, insect sex, and their possible interactions as the main factors in the analysis. Model diagnostic plots showed no violation of the normality assumption of ANOVA. Tukey HSD test was used to perform post hoc comparisons between the different treatments for each carabid species.

Mixed effects models using the function "lmer" were used to compare the number of seeds consumed by each carabid beetle from each seed species under sensory treatments[97]. Data were analyzed and presented for each species separately as significant differences between carabid species were detected in the full data set (all three carabid species together). For each predatory species, the analysis was initiated by fitting a maximal model to the data including weed species, sensory treatment, and insect sex and their possible interactions as the main effects. Replicate was used as a random blocking factor in the model to account for spatial structure in the experiment (three seed species placed into each Petri dish).

Mixed effects models were also used to compare peak areas (GC-MS chromatograms) of the chemical compounds identified for seed species. A maximal model including weed species, identity of chemical compound, and their possible interactions as the main effects was fitted to the data. Replicate was used as a random blocking factor as compounds were nested in weed species. Similar mixed effects modeling steps were used to compare the number of coated and uncoated seeds consumed by carabids in the two-choice seed coating experiments. A maximal model was fitted to the data including seed treatment, insect species, and their possible interactions as the main effects. Replicate was used as a random blocking factor as above (two patches of seeds nested in each Petri dish).

Model validity was checked by examining the distribution of model residuals throughout the mixed modeling steps described above[98]. The R package "LmerTest" was used to perform post hoc comparisons on the final models[99].

**Reporting summary**. Further information on research design is available in the Nature Research Reporting Summary linked to this article.

## Data availability
The datasets generated during and analyzed during the current study are available from the corresponding author upon reasonable request.

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

## Acknowledgements

We would like to thank Agriculture and Agri-Food Canada (AAFC) and the Saskatchewan Structural Sciences Center (SSSC) for providing excellent technical support. Funding was provided by a Discovery Grant awarded to C.J.W. by the Natural Sciences and Engineering Research Council of Canada. Financial support was provided to B.A.M. by a Natural Sciences and Engineering Research Council of Canada Industrial Research Chair (Grant 545088) and partner organizations Alberta Wheat Commission, Alberta Barley Commission, Alberta Canola Producers Commission, and Alberta Pulse Growers Commission during the preparation of the manuscript. K.A.A. was partially funded by the Western Grains Research Foundation (SK, Canada) and the Department of Plant Sciences (University of Saskatchewan).

## Author contributions

K.A.A. designed and conducted all the experiments, collected and analyzed all the data, and wrote the first draft of the manuscript. B.A.M. advised the chemical ecology work and revised the manuscript. S.M.P. advised the sensory experiments and revised the manuscript. C.J.W. secured funding, supervised all of the experimental work, and revised the manuscript.

## Competing interests

The authors declare no competing interests.
