## [Peer Review File · Communications Biology]

Reviewers' comments:

Reviewer #1 (Remarks to the Author):

I enjoyed reading this work. This work helps me to answer some of the questions that I have on the topic of carabid preferences and choices. In it originally and shows how the chemical compounds of seeds attract their predator. This topic is really important because there is not enough information about it. It can help us understand and also predict the ecosystem service called seed predation (which can be nowadays very important with the European program of Green Deal).

I do not understand properly some things or I have to disagree. Mostly I wrote them to the attached word document to make it easier to find (hope that it will be more practical for you too). In the introduction and also in the discussion I miss any note that it is not "just" chemical compounds that drive the preference but also some physical properties (such as size, mass, shape etc.) – these properties of the seeds were detected as a driver many times (for example work of Saska et al. 2019 or Honek et al. 2003).

In M&M:

Why did you use just some species and not all of them in all experiments? I am sure that for example, the full data set for *Amara* would be nice and it could show differences between granivorous (*Amara*) and predatory carabids (*Pterostichus*). It would also improve the paper.

How do you know that the volatiles was just the volatiles from the seeds and not from some microorganisms on the surface of the seeds?

How did you collect and clean the seeds? Did you use gloves? Did you dry the seeds before storage? Could you please provide more information about this time? Was the experiment made in the same year as the seeds were collected?

Thank you

Reviewer #2 (Remarks to the Author):

This paper aims to study the sensory biology of several species of granivorous or omnivorous ground beetles, which frequent cultivated fields. The authors therefore tackle key processes that can lead to the regulation of weed species in crops by manipulating ecological processes (here granivory). I salute the enormous amount of information and work carried out through this article as well as the originality of the data presented in it. I therefore think that all these data provide original and important knowledge both to dissect the food decision strategies of these insects but also to define nature-based control strategies of weeds.

This being clearly stated, I believe that the paper is not publishable in the form currently presented and that it requires a lot of rewriting. The first reason has to do with the astronomical amount of experiences and results that are presented. The second reason is linked to the complexity of reading the manuscript will develop these two points.

Regarding the first point, I think there are too many different experiences, each one being complex, that are presented. The reader, faced with this very large quantity of experiences which have been conducted and which are presented, very quickly gets lost in the text. This article deserves to be split into two different papers, the first focusing on highlighting, on several species of ground beetles, the relative role of different sensory organs in the detection of different species of seeds, the second being devoted to the different constituents of seeds and their role in the choice strategies. I have the feeling, reading this paper, that these results present the whole of a doctoral work. If the authors wish

to present all the results in the same paper, I think that it will be necessary to clearly define subcategories of experiments and indicate, at the end of the introduction, in a dedicated paragraph, where the different experiments are positioned and the questions they wish to answer. I therefore think that the final paragraph of the introduction (L90-99) should be developed in the logic of the paper and supported by bibliographical references). The authors indeed state a certain number of facts (e.g. ... omnivorous carabids use olfactory perception ... (L. 90) ... seed choice is driven by the perception of long chain volatiles ... (L91). .. seeds volatiles seem to encode information about the lipid content (L. 93) ...) for which no references are given.

Regarding the second point, I had a lot of trouble going into each experience that was done with regard to their presentation because to clearly understand the experiments that were carried out and to interpret the figures on this basis, I had to go back and forth complex between the results, the material and methods and the supplementary information. Furthermore, the presentation logic in the figures is complex. With regard to the first experience (manipulation of the sensory organs), the name of the treatments and what they correspond to, is not logically and clearly formulated. For example, does the "eyes" treatment correspond to "all sensory organs except the eyes are neutralized" or "" the eyes are neutralized "? Does the "maxillary palps" treatment correspond to "all sensory organs other than maxillary palps are neutralized" or "maxillary palps are neutralized"? In general, the description of sensory treatments is poorly written. It is necessary to clearly indicate which organs have been manipulated (for example the authors speak of gustatory organs in the text without indicating that there are two visibly, the maxillary and the labial palps) and respect a logic of identification of treatment which will be found in the figures (eg the "Palps" treatment correspond to individuals for whom both the maxillary and labial palps have been neutralized).

The different species used in the different experiments are not the same which can potentially be a problem. The authors should justify this point. For example, why the species *Harpalus* sp. is not used in the experiment shown in Figure 1? Why in the series of experiments with coated pellets or seeds (L. 153-169) it is not the same species which are always used. This could prevent drawing general conclusions. Regarding this same series of experiments, a general table presenting all the treatments that have been carried out would be of great use to help the reader understand everything that has been done and guide him in reading the results and figures.

This last remark is even more important with regard to the series of experiments which were carried out to test the role of the protein quality and the protein-lipid ratio in the seeds. Here again, it is necessary to clearly qualify the treatments and indicate, for example, that the "protein-biased" treatment corresponds to 21:21 versus 35: 7 in the choice experiments carried out. In these experiments, the presentation of geometric framework bivariate analysis is too insufficient to be able to clearly read what the graphs present.

To conclude on something positive, and I think there is plenty of material for this, this paper presents a set of very important experiences for understanding decision-making strategies in insect species that have been identified as key potentials regulating actors of weeds in agricultural fields. But I think a lot of work is needed to make the manuscript readable for the reader.

Reviewer #3 (Remarks to the Author):

Ms title: The interplay between seed volatiles and lipids drives seed choice in ground beetles

Ms #: COMMSBIO-21-3313

This paper demonstrates that seed preference of omnivorous ground (carabid) beetles is linked to the long chain volatile chemicals derived from the epicuticular lipids on the seed coat surface, leading to assessment of the fatty acid content of the seed and accurate seed choice. Seeds of three brassicaceous species (*Brassica napus*, *Sinapis arvensis*, *Thlaspi arvense*) and adults of four carabid species *Poecilus corvus*, *Harpalus amputatus*, *Pterostichus melanarius*, and *Amara littoralis* were used in the study. Additionally, accurate seed choice was demonstrated to occur through the antennae and

palps. Overall, I found the paper interesting; however, I will have various points to be addressed/considered.

1. Title: I found the title inappropriate for a research paper as it sounds like a title of a review paper. I recommend authors to come up with a more specific & conclusive & definitive title.

2. Abstract: Abstract includes general statements and does not include the details of the study. So rather than mentioning common statements, authors should highlight their specific data as the abstract should stand by itself. The tested species and seed species should be also mentioned.

Line 18: "...numerous species are predators of various pests and weed seeds". It is not accurate to make a definition as "predator of weed seeds". I recommend smthng like ".....numerous species are predators of various pests and feed on weed seeds at the same time".

3. Introduction:

Line 31: Pls replace parasite with parasitoid.

4. Results:

Line 188: "...different landscapes in both species as the lipid intake..." Which both species?

Lines 190-191: "Together, these observations suggest that this carabid species is likely lipid-limited and modulate their protein intake to address this limitation." How can protein intake be modulated to compensate lipid limitation? Pls explain or revise or omit.

5. Discussion:

Lines 253-254: "Maxillary and labial palps appear to carry significantly fewer of those chemoreceptors, as predators carrying one pair of either palps failed to make accurate seed choices". Is this interpretation correct? If the palps possess significantly fewer chemoreceptors, how come predators with one pair of either palps can fail to make accurate seed choice? Pls explain or revise or omit.

Lines 256-257: "Olfactory receptors can also be found on maxillary and labial palps, but their abundance on these appendages is usually rather low". This was already mentioned in Lines 253-254. Please avoid of repetitions.

Lines 259-261: "Nonetheless, some authors suggest that sensory information perceived through olfactory receptors is usually more accurate and specific than information perceived via gustatory receptors 55". You say some authors but cite a single paper.

Lines 268-269: "The fatty-acid-derived seed cues are located on seed surfaces and comprise three main groups of long-chain aliphatic lipids: alkanes, esters, ketones". Reference needed.

Lines 292-294: "Based on our data we suggest that seed surface lipids provide the "kairomonal" signals necessary for interactions between carabid predators and seeds of weed species to initiate and unfold 60". You make a suggestion and cite a paper. Is this your suggestion or Ref 60s? Please revise.

Lines 335-342: "The theme/idea here was already mentioned in lines 309-319. So please avoid of repetition.

6. Material and Methods:

- Section on "Elucidating the sensory mechanisms of seed detection and discrimination in carabid": How many beetles were used per tray?

Line 405: "...ablating one or more of the sensory appendages with precision tweezer". The appendages should be specified.

- Section on "Isolation, identification, and behavioral testing of seed cues that mediate seed perception in carabids": Lines 441-442: "The impact on seed foraging behaviors of seed volatile cues identified above was tested via coating protein pellets with surface extracts of different seed species.". Is there a reference for this method or this is a novel methodology introduced here in this paper?

Line 442: "The use of protein pellets (100% shrimp protein and no fat)...". How was a protein pellet without any fat was prepared from Shrimp or was it purchased?

Lines 443-445: "However, protein pellets offered a simple and homogenous (physically and chemically) food source to measure carabid feeding responses to seed surface extracts under controlled conditions." Reference needed.

The interplay between seed volatiles and lipids drives seed choice in ground beetles

Khaldoun A. Ali^{1*} Boyd A. Mori² Sean M. Prager¹ and Christian J. Willenborg¹

¹ Plant Sciences Department, College of Agriculture and Bioresources, University of Saskatchewan, Saskatoon, Saskatchewan, Canada

² Department of Agricultural, Food and Nutritional Science, University of Alberta, Edmonton, Alberta, Canada

* Corresponding author: Department of Plant Sciences, University of Saskatchewan, Saskatoon, SK, Canada S7N 5A8 (chris.willenborg@usask.ca)

Keywords: Carabidae, agroecology, weed biological control, ecosystem service, seed preference, seed volatiles, seed nutrients, seed defensive chemicals.

Abstract

Ground (~~carabid~~) beetles (Coleoptera: Carabidae) are among the most prevalent biological agents in temperate agroecosystems as numerous species are predators of various pests and weed seeds. It remains unclear why omnivorous carabids choose to include seed species in their diets when alternative food sources are available, or even why they prefer to consume seeds of one species over another. Here, we show that omnivorous carabid predators rely on olfactory perception to detect seeds of different species and identify the suitable seed species among them. Carabids exploit long chain volatile chemicals derived from the epicuticular lipids located on the seed coat surface to assess the fatty acid content of the seed. Seeds that contain desirable lipids are preferably chosen for consumption given that seed species in the environment show no significant differences in their physical characteristics. This lipid-driven choice of seeds probably helps carabids overcome some of the lipid limitations in their diets.

Commented [A1]: If I understand it well it says that seeds do not have differences in physical characteristics but what about size, shape etc.? It differs even between the seeds of one species.

Introduction

Biological control (biocontrol) is an important service provided by insects in both natural and managed ecosystems.¹ In many cases, biocontrol agents (i.e. natural enemies), including insect predators and parasites, are endemic in agricultural fields and thus offer natural control services.² In agroecosystems, natural control services help to maintain ecosystem balance and reduce reliance on pesticide inputs, which in turn helps to mitigate

pesticide resistance problems.³ Reaping such bio-ecological benefits is not an easy task to achieve in modern
agroecosystems. To determine and promote the beneficial services that natural enemies provide in the ecosystem,
it is essential to understand the basic interactions between insect predators or parasitoids, and the organisms they
target in the field.^{4,5} One of the first steps is to understand how insect predators and parasitoids perceive their
target organisms and assess their suitability for feeding or oviposition to better predict the efficacy of these natural
biocontrol agents under realistic situations.^{4,6} Such knowledge becomes even more crucial when the natural
agents of interest are generalists or omnivorous, able to feed on or oviposit into a wide array of organisms.^{4,7}
Therefore, it is vital to tease out the factors that render certain species more prone to elevated predation or
parasitism risks, especially when other species that may serve as alternative food sources to the biocontrol agent
are accessible.⁸ By understanding these core ecological aspects of feeding habits in natural enemies,
agroecosystems can be managed to improve the diversity and abundance of insect predators and parasitoids to
enhance their ecological functioning in agroecosystems.

Ground (carabid) beetles (Coleoptera: Carabidae) are one of the most important groups of predatory insects
in temperate ecosystems as they function as epigeic polyphagous predators.⁹ Carabid activity can be diurnal or
nocturnal depending on their species identity and/or habitat properties, and they are generally voracious feeders
able to consume close to their body weight of food each day.^{10,11} Numerous carabid species prey upon various
agricultural pests such as aphids,¹² lepidopteran caterpillars,¹³ dipteran eggs and midges,¹⁴ wireworms,¹⁵ and
slugs.¹⁶ In addition to pests, numerous species of predatory carabids also feed on seeds of different weed species
after seed ~~dispersion shed~~¹⁷; thus, the majority of carabid predators are omnivorous. These diverse feeding habits
in carabids make them amongst the most formidable predators in the agroecosystem as they have potential to
regulate populations of various pests and weeds.^{18,19}

Despite the high potential of omnivorous carabids as versatile biocontrol agents in agroecosystems, their
complex feeding habits make it difficult to predict their biocontrol efficiency against pest or seed species. Seed
feeding in particular remains poorly understood, and it is unclear why seed feeding habits arose given the
abundance of prey in arable fields.^{20,21} Therefore, it is difficult to predict which species of seeds or prey would
be prone to elevated carabid attacks under realistic situations. Two hypotheses have been proposed to explain
carabid seed feeding: 1) omnivorous carabids seek seed consumption only when prey species are scarce or
inaccessible; or 2) omnivorous carabids mix both food types because prey feeding alone is insufficient for survival
and development.^{8,22} It was first thought that seed feeding would be biologically important to only a small group
of granivorous carabids species that subsist on diet composed mainly of seeds of certain species.^{23,24} Beyond that,
seed feeding habits in omnivorous carabids would be opportunistic and tend to arise mostly when alternative
foods are scarce or hard to obtain.⁸ Seed feeding habits in carabids tend to actually transcend the artificial limits

Commented [A2]: There are some strictly granivorous carabid

imposed by the dietary specialization reasoning (omnivory vs. granivory) as seeds are featured in the diets of a
large segment of carabid taxa.^{25,26} This evidence suggests that seed feeding habits in carabid species arose due to
yet unexplored biological needs that are not necessarily exclusive to granivorous carabids *sensu stricto*.
Therefore, it is vital to explore the ecology of seed feeding habits in omnivorous carabids and elucidate their
impact on the ecological functioning of carabids in agroecosystems.

Seed feeding in omnivorous carabids is likely driven by behaviors that are directed towards addressing specific
biological needs. Random seed encounters were assumed to be the sole driver of seed feeding and seed selection
decisions in carabids;⁸ however, field and laboratory studies have demonstrated that carabid predators show active
selection of specific seed species when seeds of different species are accessible.^{27,28,29} Seed species preference
requires carabids to discriminate among seeds of different species, assess their suitability aspects, and then
identify the species suitable for consumption.³⁰ To perform these tasks, carabids need to collect reliable seed-
derived information through their different sensory systems.³¹ Our current understanding of the sensory and
behavioral mechanisms that underlie discrimination of, and preference for, seeds in carabids remains rather
rudimentary.³² Seed odors can influence carabid orientation responses in olfactometers,^{33,34} but it remains
uncertain whether chemoperception alone can drive accurate decision making among seed choices. It is also
difficult to ascertain whether seed feeding replaces or complements prey feeding in omnivorous carabids in terms
of fulfilling the needs for survival and development.^{22,35} Therefore, it is essential to study the sensory biology of
seed detection and discrimination in omnivorous carabids to better understand how they are able to distinguish
among seed of different species and select the suitable seed species among them. This is expected to further our
knowledge around the ecological functioning of omnivorous carabids in agroecosystems as it may help us
decipher the ecological conditions under which seed feeding may become a priority.

Here, we explore the ecology of ~~omnivorous~~ carabid beetles and describe aspects of the sensory, behavioural,
and nutritional ecology involved in seed feeding of *Poecilus corvus*, *Pterostichus melanarius*, *Harpalus*
*amputatus*, and *Amara littoralis* (Coleoptera: Carabidae). Sensory manipulation protocols and multiple-choice
seed feeding bioassays show that omnivorous carabids use olfactory perception to detect seeds of different species
and identify the suitable seed species among them. Seed choice is driven by the perception of long chain volatile
chemicals derived from the epicuticular lipids located on the seed coat surface. Seed volatiles seem to encode
information about the lipid content of seed species. Experiments with synthetic diets suggest omnivorous ground
beetles potentially consume seeds to overcome the scarcity of some fatty acids in their diets. Carabids most likely
choose seed species based on desirable lipid content provided that seed species in the environment show no
significant differences in their physical properties. Seed feeding habits can complement prey feeding in carabid
omnivores. Plant seeds are therefore salient elements of the general feeding habits of omnivorous carabids, which

Commented [A3]: I have to disagree for example Amara is usually referred as granivorous and Pterostichus as predator (for example Kulkarni et al. 2013 or Homburg 2013)

potentially places the contribution of carabids to weed biocontrol in agroecosystems on par with their pest
biocontrol services.

Results

Olfactory chemoreceptors enable carabid seed predators to detect and discriminate among seed species

Seeds of three brassicaceous species (Brassicaceae), similar in shape and color and all are considered high in
lipids, were offered to three species of sensory-manipulated carabid predators (see Methods section). Total seed
consumption was used as a measurement of carabid seed detection success under sensory manipulation
treatments. Analysis of variance showed that disabling different sensory organs significantly affected the ability
of carabid species to detect seeds in feeding bioassays ($F_{7,530} = 2.26$, $P < 0.03$; $n = 65$, Figure 1 a-c). There were
no significant differences in the responses of carabids based on sex ($F_{1,530} = 0.32$, $P = 0.56$, $n = 65$). Within
species, covering the compound eyes in *P. corvus* caused a significant reduction (ca. 30%) in seed detection
success compared to intact insects ($F_{8,207} = 17.9$, $P < 0.001$, $n = 25$; Figure 1 a). Carabids with only functional
eyes largely failed to find seeds as their seed consumption was not significantly different from zero. Carabids left
with intact antennae and/or maxillary palps (olfactory organs) found seeds with considerable success, with
consumption rates between 50-70% compared to intact beetles. Intriguingly, when predators were left with only
intact labial palps (gustatory organs), seed detection success was not statistically different from zero. Similar
response patterns were observed for both *P. melanarius* ($F_{8,206} = 19.5$, $P < 0.001$, $n = 25$; Figure 1 b), and *A.*
*littoralis* ($F_{8,117} = 16.41$, $P < 0.001$, $n = 15$; Figure 1 c). Antennae and palps in both species accounted for 50-
75% of seed detection success while those with eyes alone were not successful.

Comparing seed choice responses via mixed effects models revealed significant impacts of sensory
manipulations on seed selection decisions in the three carabid species under study ($F_{16,1552} = 2.27$, $P < 0.001$, $n =$
65; Figure 1 d-f). Responses did not differ between males and female in any of the carabid species tested ($F_{1,1552}$
= 0.01, $P = 0.9$, $n = 65$). There was a significant interaction between sensory treatments and carabid species
($F_{16,1552} = 2.22$, $P = 0.003$, $n = 65$). *P. ocellus* *corvus* showed a strong preference for *B. napus* seeds, and this
preference was maintained across all treatments where beetles had two antennae and/or four palps ($F_{8,621} = 15.16$,
$P < 0.0001$, $n = 25$; Figure 1 d). When antennae were ablated, beetles needed all four palps to make an accurate
seed choice. Accurate seed choice was lost when beetles were left with only one pair of either palps since
preference for *B. napus* seeds lost statistical significance. Accurate seed choice was also lost when antennae and
palps were ablated, leaving the beetles with functional eyes only. Similar response patterns were observed for

Commented [A4]: Not at all. The shape – Thlaspi is disk others are globular. Please remove this

both *P. melanarius* ($F_{8,621} = 26.99$, $P < 0.0001$, $n = 25$; Figure 1 e), and *A. littoralis* ($F_{8,336} = 15.88$, $P < 0.0001$, n
 $= 15$; Figure 1 f, see also Table 3 in Supplementary Information). Antennae and/or both types of palps enabled
 beetles of both species to make an accurate seed choice compared to the positive control, but the two species
 showed clear differences in their seed preference. *P. ferostichus melanarius* had a strong preference for *B. napus*
 seeds, whereas *A. littoralis* preferred seeds of *T. arvense*.

Figure 1.

The species-specific cues necessary for seed discrimination are located on the seed coat surface

The previous experiments have established that carabid seed predators rely on their chemoreceptors to detect
seeds of different species and choose the most preferable seed species among them. We originally attempted
sampling the headspace of the three brassicaceous seed species via solid phase microextraction (SPME) fibers or
dynamic air entrainment using Porpak Q, but this failed to detect the seed volatiles necessary for seed
discrimination in carabids (KA, unpublished data; also see Supplementary Information). Direct extraction of seed
surface chemicals yielded the candidate species-specific seed volatile chemicals necessary for seed discrimination
(Figure 2 a-c). Seed volatiles were composed of fatty acid derivatives comprising three main groups of long-
chain aliphatic lipids: alkanes, esters, ketones. Weed species showed significant differences in their profiles of
volatile chemicals (Mixed Effects: $F_{3,72} = 17.6$, $n = 5$, $P < 0.0001$). *Brassica napus* seeds featured the simplest
profile of surface chemistry, with only two major compounds in their profile (Table 1). By contrast, surface
chemistries of *S. arvensis* and *T. arvense* seeds showed more complex profiles of alkanes, ketones, and esters.
Fatty acid ethyl esters were not commercially available and hence further research is needed to confirm their
structure.

Chemical coating of protein pellets with seed surface chemicals revealed that seed surface chemicals stimulate
carabid feeding responses (Mixed Effects: $F_{3,100} = 15.15$, $P < 0.0001$, $n = 30$; Figure 3 a & b). Protein pellets
coated with hexane only (no seed surface chemicals) were always more preferable to *H. amputatus*, *P.*
*melanarius*, and *P. corvus* than pellets coated with *B. napus* extracts in two-choice feeding experiments (Figure
3 a). By contrast, pellets coated with *B. napus* extracts were more preferable to *P. melanarius*, *P. corvus*, and *H.*
*amputatus* when those were offered against pellets coated with *T. arvense* chemicals (Figure 3 b). Furthermore,
pellets coated with *B. napus* chemicals were the most preferable for *P. corvus*, *H. amputatus*, and *A. littoralis*
when offered against pellets coated with *S. arvensis* and *T. arvense* chemicals in three-choice feeding experiments
(Mixed Effects: $F_{2,63} = 52.45$, $P < 0.0001$, $n = 30$; Figure 3 c). These results suggest that carabids, when
determining preferable seed types, seem to choose the seed type of the simplest chemistry when all else is
physically equal.

This simple rule seems to hold under realistic situations as well. Seeds of *B. napus* that carabid predators
usually prefer became significantly less preferable to *P. corvus*, *H. amputatus*, and *A. littoralis* when coated with
*T. arvense* chemicals (Mixed Effects: $F_{2,48} = 5.26$, $P < 0.01$, $n = 30$; Figure 3 d). On the contrary, coating *T.*
*arvense* seeds with *B. napus* chemicals made them considerably more acceptable to *P. corvus*, *H. amputatus*, and
*A. littoralis*, but this change in preference was not statistically significant (Mixed Effects: $F_{2,48} = 0.04$, $P = 0.89$,
$n = 30$; Figure 3 e).

Commented [A5]: For me, this sentence is not necessary.
Or it should be cited and in the introduction.

**Figure 2.**

**Figure 3.**

**Seed preferences likely arise to help carabid predators overcome fatty acids limitations in their diets**

Synthetic diets were used to study how the dynamics of lipid and protein intake in carabid seed predators may
 impact seed preferences. *Poecilus corvius* consumed diets that were strongly protein-biased under protein-biased
 conditions (Mixed Effects: $F_{2,62} = 13.36$, $P < 0.001$, $n = 14$; Figure 4 a & b). By contrast, the dietary intake shifted
 towards lipid bias under the lipid-biased conditions. The intake target approached balance under balanced
 conditions, but remained protein-biased. Overall, lipid ingestion was tightly regulated across the different
 landscapes in both species as the lipid intake target was relatively stable across treatments. Protein intake, on the
 other hand, showed significant and strong changes across the nutritional landscapes created in the experiment.
 Together, these observations suggest that this carabid species is likely lipid-limited and modulate their protein
 intake to address this limitation. Similar response patterns were observed for *P. melanarius* (Mixed Effects: $F_{2,59}$

Commented [A6]: Do not understand the meaning. Do you mean "nutrient classes"(carbohydrates, fats etc?)

= 27.45, $P < 0.001$, $n = 14$; Figure 4 c & d). *Harpalus amputatus* also showed a tight regulation of their lipid
 intake across the different P:L conditions in the experiment (Mixed Effects: $F_{2,40} = 4.24$, $P < 0.01$, $n = 12$; Figure
 4 e & f). Unlike the other two species, however, *H. amputatus* did not have a sharp increase in protein ingestion
 when dietary lipid was scarce. Rather, this species ingested significantly more lipids under unbalanced P:L
 conditions. *Harpalus amputatus* therefore seems to employ different rules of compromise under nutritional
 imbalance compared to the other two species.

Commented [A7]: The P:L is not necessary for the plot (A, C, E) if there is a color and legend. The same suggestion in fig.6

 **Figure 4.**

 Protein quality in the diet was rendered artificially low by substituting the casein-based protein mixture with
 zein, a corn-based storage protein of low quality. Compared to a normal diet, diets with low-quality protein
 triggered a significant drop in ingestion of both protein (65-70%) and lipid (ca. 50%) in *P. corvus* (Mixed Effects:
 $F_{2,53} = 31.61$, $P < 0.001$, $n = 14$; Figure 5 a & b), and also *P. melanarius* (Mixed Effects: $F_{2,60} = 9.43$, $P < 0.001$,
 $n = 14$; Figure 5 c & d). This brought about a significant shift in intake targets towards lipid bias across treatments.
 The drop in protein intake was less severe in *H. amputatus* (ca. 25%), and was stabilized around a specific level
 (ca. 40 mg) across the three P:L conditions (Mixed Effects: $F_{2,43} = 36.19$, $P < 0.001$, $n = 12$; Figure 5 e & f). In

all three species, lipid ingestion was the lowest when the P:L ratio in the diet was highest. Lipid intake showed a
 progressive and significant increase as P:L ratio fell and moved towards lipid bias.

 **Figure 5.**

 When the diet was laced with the seed toxin allyl isothiocyanate at 0.5% (v/v) without manipulating protein
 quality, similar but weaker protein avoidance (40-50%) and shift towards lipid-biased intake targets were
 observed in *P. corvus* (Mixed Effects: $F_{2,60} = 23.4$, $P < 0.001$, $n = 14$; Figure 6 a & b), and *P. melanarius* (Mixed
 Effects: $F_{2,60} = 9.43$, $P < 0.001$, $n = 14$; Figure 6 c & d). Despite the drop in protein intake, lipid intake targets in
 both species were not significantly different from the normal diet. By contrast, *H. amputatus* did not reduce its
 protein intake in the presence of allyl isothiocyanate, but increased lipid intake by almost two-fold (Mixed
 Effects: $F_{2,47} = 18.39$, $P < 0.001$, $n = 12$; Figure 6 e & f). Adding allyl isothiocyanate to the diet at 2.5% (v/v)
 caused carabids to lose the ability to self-compose an optimal diet, suggesting there is an upper limit on carabid
 tolerance to seed toxins. Overall, nutrient intake in carabid seed predators was a complex process influenced not
 only by nutrient availability, but also aspects of food quality (see Table 3 in Supplementary Information).

**Figure 6.**

**Discussion**

Our study has demonstrated that omnivorous carabid seed predators rely mainly on chemoperception to detect
 and choose among different species of seeds. The sensory information needed for these tasks are encoded in
 volatile chemicals located on the seed surface and is detected by the chemoreceptors located on the antennae and
 palps. Chemoperception has been reported to guide essential aspects of prey searching and detection in
 omnivorous and carnivorous carabids as well.^{36,37} Visual cues did not elicit the sensory response necessary to
 guide seed choice. This is sensible as plant seeds are sessile and usually scattered on the soil surface or even
 buried underneath it,³⁸ which presumably should make visual detection of seeds by carabid predators difficult,
 especially against the soil background. Similarly, visual perception can be unreliable for detection of sessile prey
 in carnivorous carabids.^{39,40} Carabid visual receptors may be more tuned towards detecting prey movement, and
 should be more useful for hunting down mobile prey than locating sessile insect prey or seeds.⁴¹ It is important

to mention here that sensory manipulations, although intrusive, did not appear to cause significant detriment to
the seed selection ability in the carabids under study. Sensory-manipulated carabids carrying sufficient functional
chemoreceptors (antennae, palps, or both) were still able to exhibit accurate seed choices akin to those of intact
insects (positive control). Sensory manipulations as such did not seem to affect the ability of carabids for
information processing or decision making. Insect sensory appendages carry receptors that collect information
from the surrounding environment, but play no major role in processing the sensory input or releasing of
behavioral responses.^{42,43} Higher cognitive centers like optic lobes, antennal lobes, lateral horns, and mushroom
bodies are responsible for processing the sensory input and releasing appropriate behavioral responses,^{44,45} and
those remained intact in the carabids under study.

The carabid species we studied carry most of the chemoreceptors responsible for seed detection on their
antennae.^{46,47,48} Antennae, either alone or with palps, enabled carabids to identify the suitable seed species with
high accuracy. Maxillary and labial palps appear to carry significantly fewer of those chemoreceptors, as
predators carrying one pair of either palps failed to make accurate seed choices. Carabid antennae usually carry
an abundance of olfactory receptors that enable predators to collect specific chemical information about their
food.^{49,50} Olfactory receptors can also be found on maxillary and labial palps, but their abundance on these
appendages is usually rather low.⁵¹ On the contrary, gustatory receptors are usually more abundant on maxillary
and labial palps.^{52,53} Olfactory and gustatory receptors show considerable similarities in their structures and
physiological functioning, and collect chemical information of similar nature.⁵⁴ Nonetheless, some authors
suggest that sensory information perceived through olfactory receptors is usually more accurate and specific than
information perceived via gustatory receptors.⁵⁵ These lines of reasoning might explain why accurate seed choice
could take place in all treatments where antennae were not ablated from carabid heads and might also explain
why all four maxillary and labial palps were needed for accurate seed choices to take place. Beetles carrying
either type of palps alone did not seem to perceive the chemical information necessary for accurate seed choices.
Given our findings olfactory receptors are, in all likelihood, the type of chemosensilla responsible for seed
perception in carabid seed predators.

We have shown that the chemical cues that enable carabid predators to select between seeds of different species
are composed of fatty acid derivatives. The fatty-acid-derived seed cues are located on seed surfaces and comprise
three main groups of long-chain aliphatic lipids: alkanes, esters, ketones. Simple profiles of surface volatiles like
*B. napus* seem more preferable than complex profiles as coating of certain pellets showed. Complex surface
chemistry seems to encode information that deters feeding. No glucosinolate compounds were detectable in
extracts of seed surface chemicals, even though seeds of brassicaceous species usually harbor considerable
amounts of these defensive compounds.⁵⁶ Other authors have also reported that glucosinolates and their

breakdown products (isocyanates) are not usually detectable in headspaces of brassicaceous species, or in extracts
of their epicuticular lipids.^{57,58} These findings contradict the belief that glucosinolates may act as deterrents
against carabid seed predation. *Amara littoralis* in our feeding multiple-choice experiments showed a strong
preference for *T. arvense* over *B. napus*. This strong preference for *T. arvense* seeds was probably not based on
seed defensive chemicals because seeds of *T. arvense* usually contain high levels of glucosinolates.⁵⁹ It could be
proposed that glucosinolates and their breakdown products are unlikely to function as preingestive seed feeding
deterrents for carabid seed predators.⁶⁰ The impact of glucosinolates on seed selection decisions does not seem to
be dichotomous. Instead, glucosinolates seem to play more complex postingestive roles as they influence nutrient
intake dynamics rather than deter feeding altogether (see below). By contrast, seed surface aliphatic lipids have
emerged as the main preingestive signaling chemicals carabid predators exploit to guide their seed foraging
efforts. It remains uncertain if the same applies for non-brassicaceous seeds.

Aliphatic lipids are an essential constituent of the cuticle layer that covers surfaces of both somatic and
reproductive plant tissues including seeds, and usually serve wide ecological functions.^{61,62} Behavioral testing of
the identified seed surface lipids via coating protein pellets or coating seeds themselves showed that seed surface
chemicals were able to drive feeding responses of carabids. These feeding stimulatory effects validates our
previous conclusion and fits into an ample body of evidence documenting plant surface lipids as interlocutors of
feeding and oviposition preferences in insects.⁶³ For instance, surface lipids have been shown to mediate essential
aspects of feeding behaviors and host plant choice in insect herbivores of Diptera,⁶⁴ Lepidoptera,⁶⁵ Coleoptera,⁶⁶
Thysanoptera,⁶⁷ Hymenoptera,⁶⁸ and Hemiptera.⁶⁹ Based on our data we suggest that seed surface lipids provide
the “kairomonal” signals necessary for interactions between carabid predators and seeds of weed species to
initiate and unfold.⁶⁰ While the kind of information carabid seed predators extract from seed kairomones remains
unknown, there are well-documented (yet poorly understood) correlations between plant species identity, cellular
fatty acid metabolic-biosynthetic pathways, and composition of seed surface lipids.^{62,70} It is quite possible that
seed surface lipids carry information about the fatty acid composition of the seed, and potentially also of their
quantity or quality. The information encoded in seed surface lipids appears vital for carabid interactions with
seeds as such interactions could not take place when carabid predators were stripped of their olfactory
chemosensilla. Based on this, we propose that carabid beetles employ olfactory templates or search images to
guide their seed feeding behaviors. It remains to be explored if the olfactory templates that guide seed recognition
and choice are hardwired in carabid brains or formed through non-associative or associative learning
mechanisms.⁷¹ Olfactory priming of carabid seed predators with odors of specific seed species was found to not
affect seed choice responses in carabids as a non-associative learning mechanism (KA, *unpublished data*).

Perhaps more sophisticated mechanisms of associative learning mediate the formation of the olfactory templates
that guide seed selection decisions in carabid seed predators.

Carabids tested in this study showed a tight regulation of their lipid intake when dietary protein-to-lipid ratios
were out of balance. Lipids appear to be limiting to nutrient foraging in carabids may be because they are less
accessible or more difficult to obtain compared to protein. This is reasonable given that prey species in arable
fields are usually deficient in essential lipids.²⁰ If carabid predators feed on prey alone, they need to consume
excessive amounts of prey to extract sufficient lipids for survival and development.⁷² Different extents of
excessive protein ingestion were observed for the carabid species tested, suggesting different levels of carabid
adaption to lipid scarcity in their diets. However, the excessive intake of protein is usually deleterious to
predators' survival.⁷³ Feeding on prey alone would be suboptimal for carabid feeding ecology. Carabid foraging
strategies that feature seed feeding habits might therefore be driven by the need to acquire essential lipids that are
scarce in alternative food sources. This is plausible given that some species of carabid predators collected from
arable fields have been reported to suffer lipid limitations in their diets.^{74,75} Scarcity of lipids as might be among
the major nutritional challenges that carabids seed predators need to surmount in order to survive, and more
research is needed to verify the generality of this phenomenon.

The tested carabid species also adjusted their nutrient foraging decisions in response to different parameters
of diet quality. When protein quality in our synthetic diets was made low, strong protein avoidance responses
were triggered and nutrient intake shifted towards lipid bias as protein intake dropped significantly relative to
lipids across all three carabid species. Similar avoidance of low-quality protein has also been reported for other
insect species as it is detrimental to insect survival in general.^{76,77} Quality more than quantity of dietary protein
imposes strict limits on lipid foraging in carabid seed predators. Presence of allyl isothiocyanate in carabid diets
did not deter feeding, but affected regulation of nutrient intake in complex ways. Allyl isothiocyanate triggered
protein avoidance in both *P. corvus* and *P. melanarius* as these two species reduced their protein intake by almost
half without reducing lipid intake compared to a normal diet. Allyl isothiocyanate is among the secondary
metabolites that binds to protein and reduces its quality by rendering it less digestible by insects.^{76,78} Intriguingly,
this effect did not cause *H. amputatus* to avoid protein, but rather to significantly increase its lipid intake, perhaps
to account for high detoxication costs.⁷⁹ Nevertheless, allyl isothiocyanate at tolerable levels did not strongly
constrain our carabid seed predators from reaching their lipid intake targets. Seed toxins therefore might not
always confer protection against carabid predation as carabids seem to prioritize lipid acquisition when seeds are
chemically defended.

Mixed feeding habits that combine seeds and prey are likely more optimal for carabids to overcome the dietary
challenges they encounter in their environments.^{80,81} In this way, carabids can obtain scarce lipids, acquire

nutritious proteins, eschew harmful proteins, and avoid the detriment of protein overconsumption. Mixed feeding
could also dilute seed defensive chemicals and mitigate their harmful effects.^{82,83} This could explain why a large
proportion of carabid taxa are omnivorous and tend to include nontrivial amounts of seeds in their diets while
true granivores remain rare. It might also explain why lipid-rich seeds are chosen more preferably for
consumption by carabids when all else is physically equal,^{84,85} and why carabids maintain their strong preference
for weed seeds even when prey is offered as an alternative food alongside seeds.^{86,87} This could be why true
specialized seed feeding habits remain rare in carabids and restricted to certain environments where an abundance
of certain seeds is maintained frequently enough to allow physiological adaptation.⁸⁸ We only tested the possible
effects of seed chemistry on carabid seed choice in this study. Hence, the conclusions reached here cannot be
extended to cases where seed species show considerable differences in their physical characteristics.⁸⁹ Based on
the results presented here, when examined along with previously published studies,⁸⁴⁻⁸⁷ we can conclude lipid-
rich seeds are more likely to suffer elevated carabid attacks in arable lands. Albeit within certain limits of protein
quality, seed toxicity, and physical seed characteristics. We further conclude that these choices are mediated by
carabid chemoreceptors detecting seed-derived volatile cues. Finally, we conclude that the chemistry-driven seed
feeding habits in omnivorous carabid beetles complements rather than replaces prey feeding. Plant seeds could
be salient elements of the general feeding habits of omnivorous carabids, and their contribution to weed biocontrol
in agroecosystems should thus be nontrivial even if prey is abundant. This corroborates the notion that
omnivorous carabids are among the formidable versatile predators in agroecosystems, and their weed biocontrol
services are most likely on par with their pest biocontrol services.

**Materials and Methods**

*Seed material*

Seeds of three different brassicaceous species (Brassicaceae: *Brassica napus* L., *Sinapis arvensis* L., *Thlaspi*
*arvense* L.) were used as model species in this study. Seeds of these three species were similar in size, shape,
color, and surface texture, all were considered high in lipids, and all are weeds of considerable importance in
arable fields of the Northern Great Plains region of North America.²⁷ Previous work has shown that each of the
chosen weed species represented a seed type of a specific acceptability rank to carabid seed predators.^{27,32}
Accordingly, seeds of canola (*B. napus*) were used as a highly preferable seed type, whereas seeds of wild mustard
(*S. arvensis*), and field pennycress (*T. arvense*) represented moderately and weakly acceptable seed types,
respectively. Seed masses of the three weed species were obtained from stored samples (5 °C) collected in

Commented [A8]: As it is written above it is not true, or if you think so can you make some table where it will be demonstrated? Please remove it and mention in the discussion that the physical properties can differ and change the preferences.

summers of 2016-17. Seeds were collected from different field sites at the Kernen Crop Research Farm near
Saskatoon, SK, Canada (52°09'10.3" N 106°32'41.5" W).

*Carabids*

Adults of the omnivorous carabid species *Poecilus corvus* (Leconte), *Harpalus amputatus* Say, *Pterostichus*
*melanarius* (Illiger), and *Amara littoralis* Dejean which are known to consume weed seeds, were used in this
study. Live adults were collected from different field sites at the Kernen Crop Research Farm in the summers of
2018-20 via dry pitfall trapping. Field sites chosen for carabid trapping were seeded with canola, pulse, or cereal
crops. Pitfall traps consisted of two plastic 0.5 L cups (10 cm height × 8 cm diameter), one acted as a sleeve and
was buried into the soil and kept flush with the soil surface, and the other cup (the actual trap) was inserted into
it.⁹⁰ Pitfall traps were enclosed into cages of fine wire mesh ($\sigma = 1.1$ cm) to prevent vertebrates from entering the
traps and ravaging the catches. Traps were emptied every three days and the collected insects were placed into
plastic boxes (40 cm × 25 cm, 25 cm depth) lined with plant material and moist filter paper then brought to the
laboratory for identification and experimentation. Carabid species identity and sex of the experimental carabids
were determined using keys in Lindroth (1961-1969).⁹¹

*Elucidating the sensory mechanisms of seed detection and discrimination in carabids*

Seeds of the three brassicaceous weed species were offered to carabid species in multiple-choice feeding
bioassays. Feeding arenas were made from a large Petri dish ($\varnothing = 25$ cm, 5 cm depth) lined with a 2-cm layer of
sterilized, moist sand as a neutral and easy-to-sterilize substrate. Seeds were placed into plastic tray rings ($\varnothing =$
28 mm, 6 mm depth) filled with white plasticine and then placed near the perimeter of the Petri dish. Plasticine
has been shown not to interfere with seed preference in carabid seed predators.⁹² A total of three trays each
harboring 25 seeds of one species were placed into each Petri dish so that the seed patch was level with the sand
layer. Imbibed seeds were used for all the feeding experiments. Seeds were imbibed by placing seed masses on
wet filter paper in Petri dishes ($\varnothing = 6$ cm, 2 cm depth), and leaving seeds to absorb moisture for 24 h in a growth
chamber at 21 ± 1 °C.²²

After collection, beetles were starved for 72 h prior to feeding experiments to empty their guts and standardize
their hunger level.³⁴ Beetle starvation was carried out by placing a single beetle (to prevent cannibalism) into a
clean and sterile Petri dish ($\varnothing = 6$ cm, 2 cm depth) lined with a moist filter paper. Petri dishes were then incubated
in a growth chamber at 21 ± 1 °C and 16:8 L:H photoperiod.⁹³ The 72-hour period was also sufficient for any
olfactory memory that might have formed while beetles had been foraging in the field to decay.⁹⁴ After 72 hours

Commented [A9]: Not with preference but it can have changed the volatile (I just guess because the plasteline is usually fatty). Have you tested it? Can you write which plasteline did you use there are many types of it?.

Commented [A10]: L:D period – Light: Dark? I am not sure what H should mean here

of starvation, the Petri dishes were placed in a refrigerator at 5 °C for 20 min to reduce the activity of the carabid
predators prior to sensory treatment.

Sensory treatments were carried out by covering the compound eyes with permanent black ink and/or ablating
one or more of the sensory appendages with precision tweezers under a stereoscope. Ablation of insect sensory
appendages is widely used for sensory perception studies.^{95,96} Ablation enabled the creation of different groups
of carabid beetles each lacking the ability to perceive sensory information of a specific nature (i.e. visual,
olfactory, or gustatory). Seed feeding responses of the sensory-manipulated carabids were then compared to three
control groups: positive, unilateral, and negative. The positive control represented ‘intact’ beetles with a full
complement of functional sensory organs. Beetles in the unilateral control had their sensory organs ablated and
the compound eye covered only on one side of the head. The side on which the sensory treatment was carried out
(left or right) was randomized to avoid bias. Finally, the negative control group was created by ablating all of the
sensory organs on the carabid head and blackening both compound eyes.

Sensory-manipulated carabids were given 10 min to ‘acclimatize’ and were then released into the feeding
arenas. Feeding arenas were incubated in a growth chamber at 21±1°C, and 16:8 L:D photoperiod and beetles
were left to feed for 5 consecutive days (Petit et al., 2014) ⁹⁰. At the end of the experiment, beetles were removed
and the number of seeds consumed from each seed patch was recorded. Each beetle was used only once and all
treatments and controls were replicated 25 times for both *P. corvus* and *P. melanarius*, and 15 times for *A.*
*littoralis*. The sex ratio was close to 50♂:50♀ across all treatments and controls.

*Isolation, identification, and behavioral testing of seed cues that mediate seed perception in carabids*

Seed surface chemicals were extracted by placing 500 mg ~~masses~~ of imbibed seeds in clean and sterile 5 ml
glass tubes. Following this, 3 ml of a 9:1 mixture of *n*-hexanes: di-chloromethane (non-polar and polar solvents
respectively) was added to each seed mass and shaken thoroughly for 15 minutes.⁹⁷ Preparations were then sealed
with parafilm and incubated in a growth chamber at 21±1°C for 72 hours. After incubation, the solvent mixture
was removed and placed into a new clean, sterile glass tube. All extracts were completely dried under a gentle
stream of nitrogen, then re-eluted into 200 µl of *n*-hexanes,⁹⁷ and then stored at -80 °C until gas chromatography
– mass spectrometry (GC-MS) analysis. Five independent extracts were carried out per each seed species, and
blank extracts (no seed added) served as negative controls.

Seed chemical extracts were analyzed by the GC-MS to identify any volatile chemical compounds isolated.
The GC-MS analysis was initiated by injecting aliquots of the volatile extracts (2 µl) into a HP-1 capillary GC
column (50 m × 0.32 mm i.d., 0.55 µm film thickness) equipped with a cool on-column injector and coupled to a

Commented [A11]: Is this method widely used?

Commented [A12]: Please change to the correct citation style.

mass spectrometer (JEOL AccuTOF 4G, USA). The GC (Agilent 7890A, USA) was programmed as follows:
oven temperature was maintained at 50 °C for 2 min and then programmed at 5 °C/min to 250 °C using helium
as carrier gas. The initial identification of any detected compounds was done by comparing retention indices
(Kovats Index) of the detected peaks to published spectrum libraries. The TSS Utility software with a link to the
NSIT Library was used for analyzing the chromatograms and identifying the detected peaks. Authentic samples
of the compounds identified by GC-MS analysis were purchased from Sigma Aldrich (Sigma Aldrich, Canada)
for structural confirmation.

The impact on seed foraging behaviors of seed volatile cues identified above was tested via coating protein
pellets with surface extracts of different seed species. The use of protein pellets (100% shrimp protein and no fat)
was not intended to completely mimic the seeds as seeds are not made up entirely of protein. However, protein
pellets offered a simple and homogenous (physically and chemically) food source to measure carabid feeding
responses to seed surface extracts under controlled conditions. Protein pellets coated with different seed surface
chemicals were used in three sets of multiple-choice feeding experiments to test the impact of seed surface
chemicals on feeding responses in carabid seed predators. Similar coating techniques were also adopted to
manipulate the surface chemistry of seed species themselves. The aim of this experiment was to test if seed
surface chemistry, and therefore their preferability to carabid predators, could be changed by the chemical coating
procedures (for full details see Supplementary Information).

*Exploring the nutritional basis of seed preference in carabids*

Synthetic diets of three different protein-to-lipid (P:L) ratios (*viz.* 35:7, 21:21, 7:35) were prepared as described
by Simpson & Abisgold (1985).⁹⁸ In brief, diets represented a complete mix of nutrients, containing fixed levels
of proteins and lipid (42% dry weight), amended with micronutrients, salts, and vitamins (4%). The remaining
54% of the diets were filled with cellulose as a non-nutritive bulking agent in order to maintain a constant bulk.
The protein sources in each diet represented a 3:1:1 mixture of casein, bacteriological peptone and egg albumen,
whereas lard (pure fat) was used as the main source of lipid.^{74,99} The prepared dry dietary mixtures, before being
presented to the carabids, were suspended in 2% agar as a 5:1 agar: dry food formula resulting in *ca.*86% water
content.⁷⁷

The impact of P:L ratio on dietary intake regulation was studied via offering three different combinations of
the P:L diets described above in two-choice feeding bioassays. In short, the feeding bioassays were set up in
Petri-dish ($\emptyset = 25$ cm, 5 cm depth) by dividing each petri dish into two arenas (two halves). In each arena, a
block of synthetic food (400 - 500 mg diet cubes) was placed near the perimeter of the dish. Each food block

represented a food source containing a known P:L ratio, and the placement of the food blocks was randomized in
order to avoid bias. Treatment groups were established as three different pairings of the P:L diet blocks: 35:7 +
7:35; 21:21 + 7:35; 21:21+ 35:7. The P:L pairings created an experimental bi-dimensional nutritional landscapes
that covered protein-to-lipid values ranging between P:L= 5:1 and P:L= 1:5. These ratios were not based on
estimates of protein and fatty acids in brassicaceous seeds. Rather, diet pairings represented a standard laboratory
protocol adopted for determining which nutrient is limiting to carabid nutrient foraging decisions of *P. corvus*,
*P. melanarius*, and *H. amputatus*. The impact of protein quality in carabid diet on nutrient intake decisions was
then tested. Protein quality in the synthetic diets was manipulated to create two levels of protein quality in the
food blocks offered to carabids. The level of protein quality was changed by replacing 100% of casein in the
normal diet with zein, a low-quality protein.⁷⁷ In another set of experiments casein was not substituted but the
normal diet was augmented with allyl isothiocyanate at 0.5% and 2.5% v/v. The aim here was to investigate how
nutrient intake decisions in carabids might change when their diets contained seed defensive chemicals.

**Statistical analysis**

R v.4.0.3 (R Development Team 2020) was used for all data analysis. Three-way analysis of variance
(ANOVA) was used to compare the total number of seeds consumed by each carabid over five days under the
different sensory treatments. Significant differences between carabid species were detected when the full data set
was analyzed, so data were analyzed and presented for each species separately. The analysis was carried out by
fitting a maximal model including sensory manipulation, insect species, insect sex, and their possible interactions
as the main factors in the analysis. Model diagnostic plots showed no violation of the normality assumption of
ANOVA. Tukey HSD test was used to perform post-hoc comparisons between the different treatments for each
carabid species.

Mixed effects models using the function “lmer” were used to compare the number of seeds consumed by each
carabid beetle from each seed species under sensory treatments.¹⁰⁰ Data were analyzed and presented for each
species separately as significant differences between carabid species were detected in the full data set (all three
carabid species together). For each predatory species, the analysis was initiated by fitting a maximal model to the
data including weed species, sensory treatment, and insect sex and their possible interactions as the main effects.
Replicate was used as a random blocking factor in the model to account for spatial nestedness in the experiment
(three seed species placed into each Petri dish).

Mixed effects models were also used to compare peak areas (GC-MS chromatograms) of the chemical
compounds identified for seed species. A maximal model including weed species, identity of chemical compound,
and their possible interactions as the main effects was fitted to the data. Replicate was used as a random blocking

factor as compounds were nested in weed species. Similar mixed effects modeling steps were used to compare
the amount of food (mg) consumed by carabids from the protein pellets offered in the in two-choice or three-
choice feeding experiments. A maximal model was fitted to the data including pellet treatment, insect species,
and their possible interactions as the main effects. Replicate was used as a random blocking factor as above (two
or three patches of pellets nested in each Petri dish). Similar modeling steps were followed to compare seed
consumption responses by carabids in response to seed coating treatments.

The amount of protein and lipid consumed by carabids (dry mg) over five days under the three different
experimental P:L conditions were compared via mixed effects modeling. The initial data analysis was carried out
on each carabid species and for each experiment separately. Analysis was initiated by fitting a maximal model to
the data including macronutrient, nutritional landscape, insect sex, body mass, and their possible interactions as
the main effects. Replicate was used as a random blocking factor (two diet blocks in each Petri dish). Protein-to-
lipid ratios in the food blocks was also included in the random term of the model. Finally, Geometric Framework
(GF) bivariate analyses were used to estimate and visualize the dietary intake targets of carabids under different
protein-to-lipid ratios and different conditions of diet quality.¹⁰¹

Model validity was checked by examining the distribution of model residuals throughout the mixed modeling
steps described above.¹⁰² The R package “LmerTest” was used to perform post-hoc comparisons on the final
models.¹⁰³

**Acknowledgements**

We would like to thank Agriculture and Agri-Food Canada (AAFC) and the Saskatchewan Structural Sciences Center
(SSSC) for providing excellent technical support. Funding was provided by a Discovery Grant awarded to CJW by the
Natural Sciences and Engineering Research Council of Canada. Financial support was provided to BAM by a Natural
Sciences and Engineering Research Council of Canada Industrial Research Chair (Grant 545088) and partner organizations
Alberta Wheat Commission, Alberta Barley Commission, Alberta Canola Producers Commission, and Alberta Pulse
Growers Commission during the preparation of the manuscript.

**Author Contributions**

KAA designed and conducted all the experiments, collected and analyzed all the data, and wrote the first draft of the
manuscript. BAM advised the chemical ecology work and revised the manuscript. SMP advised the sensory and nutritional
ecology work and revised the manuscript. CJW secured funding, supervised all of the experimental work, and revised the
manuscript.

**Conflict of interests**

Authors declare no conflict of interests.

**Data availability**

Datasets analyzed and presented in this study are available from the corresponding author upon request. Data sets will be
uploaded to a data repository and this statement will be updated once data are published and an accession number or DOI
is provided.

**References**

[revised manuscript text omitted]

- 30. Sih, A., & Christensen, B. Optimal diet theory: when does it work, and when and why does it fail? *Anim. Behav.* **61**,
379-390 (2001).
- 31. Barron, A.B., Gurney, K.N., Meah, L.F.S., Vasilaki, E., & Marshall, J.A.R. Decision-making and action selection in
insects: inspiration from vertebrate-based theories. *Front. Behav. Neurosci.* **9**, 216 (2015).
- 32. Kulkarni, S.S., Dossdall, L.M., Spence, J.R., & C.J. Willenborg, C.J. The role of ground beetles (Coleoptera:
Carabidae) in weed seed consumption: a review. *Weed. Sci.* **63**, 355-376 (2015).
- 33. Kulkarni, S.S., Dossdall, L.M., Spence, J.R., & Willenborg, C.J. Seed detection and discrimination by ground beetles
(Coleoptera: Carabidae) are associated with olfactory cues. *PLoS One.* **12**, e0170593 (2017).
- 34. Law, J.J., & Gallagher, R.S. The role of imbibition on seed selection by *Harpalus pensylvanicus*. *Appl. Soil. Ecol.*
**87**, 118-124 (2015).
- 35. Saska, P. Contrary food requirements of the larvae of two *Curtunotus* (Coleoptera: Carabidae; *Amara*) *Amara* species.
*Ann. Appl. Biol.* **147**, 139-144 (2005).
- 36. Mundy, C.A., Aleen-Williams, L.J., Underwood N., & Warrington, S. Prey selection and foraging behavior by
*Pterostichus cupreus* L. (Col., Carabidae) under laboratory conditions. *J. Appl. Entomol.* **124**, 349-358 (2000).
- 37. Thomas, R.S., Glen, D.M., & Symondson, W.O.C. Prey detection through olfaction by the soil-dwelling larvae of
the carabid predator *Pterostichus melanarius*. *Soil Biol. Biochem.* **40**, 207-216 (2008).
- 38. Dessaint, F., Chadoeuf, R., & Barrales, G. Spatial pattern analysis of weed seeds in the cultivated soil seed bank. *J.*
*Appl. Ecol.* **28**, 721-730 (1991).
- 39. Wheeler, C.P. Prey detection by some predatory Coleoptera (Carabidae and Staphylinidae). *J. Zool.* **215**, 171-185
(1989).
- 40. Oster, M., Smith, L., Beck, J.J., Howard, A., & Field, C.B. Orientational behavior of predaceous ground beetle
species in response to volatile emissions identified from yellow starthistle damaged by an invasive slug. *Arthropod-*
*Plant. Inte.* **8**, 429-437 (2014).
- 41. Srinivasan, M.V., Poteser, M., & Karl, K. Motion detection in insect orientation and navigation. *Vision Res.* **39**, 2749-
2766 (1999).
- 42. Sato, K. & Touhara, K. Insect olfaction: receptors, signal transduction, and behavior. *Cell.* **47**, 121-38 (2009).
- 43. Leal, W.S. (2013). Odorant reception in insects: roles of receptors, binding proteins, and degrading enzymes. *Ann.*
*Rev. Entomol.* **58**, 373-391 (2013).
- 44. Schmidt, H.R., & Benton, R. Molecular mechanisms of olfactory detection in insects: beyond receptors. *Open Biol.*
**10**, 200252 (2020).
- 45. Prokopy, R.J., & Owens, E.D. Visual detection of plants by herbivorous insects. *Ann. Rev. Entomol.* **28**, 337-364
(1983).
- 46. Ploomi, A. et al. Antennal sensilla in ground beetles (Coleoptera: Carabidae). *Agron. Rese.* **1**, 221-228 (2003).

- 47. Merivee, E., Martman, H., Must, A., Milius, M., Williams, I., & Mand, M. Electrophysiological responses from
neurons of antennal taste sensilla in the polyphagous predatory ground beetle *Pterostichus oblongopunctatus*
(Fabricius 1787) to plant sugars and amin acids. *J. Insect Physiol.* **54**, 1213-1219 (2008).
- 48. Merivee, E., Ploomi, A., Luik, A., Rahi, M., & Smmelseg, V. Antennal sensilla of the ground beetle *Platynus*
*dorsalis* (Pontoppidan, 1763) (Coleoptera: Carabidae). *Micros. Res. Tech.* **55**, 339-349 (2001).
- 49. Merivee, E., Ploomi, A., Rahi, M., Bresciani, J., Ravn, H.P., Luik, A., & Smmelseg, V. Antennal sensilla of the
ground beetle *Bembidion properans* Steph. (Coleoptera: Carabidae). *Micron.* **33**, 429-440 (2002).
- 50. Giglio, A., Perotta, E., Talarico, F., Brandmayr, T.E. & Ferrera, E.A. Sensilla on the maxillary and labial palps in a
helicophagous ground beetle larva (Coleoptera: Carabidae). *Acta. Zool.* **200**, 1463-6393 (2013).
- 51. Van Naters, W.V.D.G., & J.R. Carlson, J.R. Receptors and neurons for fly odors in *Drosophila*. *Curr. Biol.* **17**, 606-
612 (2007).
- 52. Amrein, H., & Throne, N. Gustatory perception and behavior in *Drosophila melanogaster*. *Curr. Biol.* **15**, R673-
R684 (2005).
- 53. Su, C.Y., Menuz, K., & Carlson, J.R. Olfactory perception: receptors, cells, and circuits. *Cell.* **139**, 45-59 (2009).
- 54. Krieger, J., & Breer, H. Olfactory receptors in invertebrates. *Science.* **286**, 720-723 (1999).
- 55. Chapman, R.F. *The Insects: Structure and Function.* 4th edn. 1-584 (Cambridge University Press, 1998).
- 56. Bhandari, S.R., Jo, J.S., & Lee, J.G. Comparisons of glucosinolate profiles in different tissues of nine Brassica crops.
*Molecules.* **20**, 15827-15841 (2015).
- 57. Reifenrath, K., Riederer, M., and Muller, M. Leaf surface wax layers of Brassicaceae lack feeding stimulants for
*Phaedon cochleariae*. *Entomol. Exp. Appl.* **115**, 41-50 (2005).
- 58. Stadler, E., & Reifenrath, K. Glucosinolates on the leaf surface perceived by insect herbivores: review of ambiguous
results and new investigations. *Phytoch. Rev.* **8**, 207-225 (2009).
- 59. Warwick, S.I., Francis, A., & Susko, D.J. The biology of Canadian weeds. 9. *Thlaspi arvense* L. (updated). *Can. J.*
*Plant. Sci.* **82**, 803-823 (2002).
- 60. Sharma, A., Sandhi, R.K., & Reddy, G.V.P. A review of interactions between insect biological control agents and
semiochemicals. *Insects.* **10**, 439 (2019).
- 61. Moyna, P., & Garcia, M. Chemical composition of oat seed epicuticular lipids. *J. Sci. Food. Agric.* **34**, 209-211
(1983).
- 62. Kunst, L., & Samuels, A.L. Biosynthesis and secretion of plant cuticular wax. *Prog. Lipid. Res.* **42**, 51-80 (2003).
- 63. Eigenbrode, S.D., & Espelie, K.E. Effects of plants epicuticular lipids on insect herbivores. *Annu. Rev. Entomol.* **40**,
171-194 (1995).
- 64. Finch, S. Volatile plant chemicals and their effect on host plant by the cabbage root fly (*Delia brassicae*). *Entomol.*
*Exp. Appl.* **24**, 350-359 (1978).
- 65. Udayagiri, S., & Mason, C.E. Epicuticular wax chemicals in *Zea mays* influence oviposition in *Ostrinia nubilalis*. *J.*
*Chem. Ecol.* **23**, 1675-1687 (1997).

- 66. Adati, T., & Matsuda, K. The effect of leaf surface wax on feeding of the strawberry leaf beetle, *Galerucella*
*vittaticollis*, with reference to host plant preference. *Tohoku. J. Agric. Res.* **50**, 57-61 (2000).
- 67. Damon, S.J., Groves, R.L., & Harvey, M.J. Variation for epicuticular waxes on onion foliage and impacts on numbers
of onion thrips. *J. Am. Soc. Hortic. Sci.* **139**, 495-501 (2014).
- 68. Braccini, C.L., Vega, A.S., Chludil, H.D., Leicach, S.R., & Fernandez, P.C. Host selection, oviposition behavior and
leaf traits in a specialist willow sawfly on species of *Salix* (Salicaceae). *Ecol. Entomol.* **38**, 617-626 (2013).
- 69. Wojcicka, A. Effects of epicuticular waxes from triticale on the feeding behaviour and mortality of the grain aphid,
*Sitobion avenae* (Fabricius) (Hemiptera: Aphididae). *J. Plant. Prot. Res.* **56**, 39-44 (2016).
- 70. Medina, E. et al. Taxonomic significance of the epicuticular wax composition in species of genus *Clusia* from
Panama. *Biochem. Syst. Ecol.* **34**, 319-326 (2006).
- 71. Webster, B., Qvarfordt, E., Olsson, U., & Glinwood, R. Different roles for innate and learnt behavioral responses to
odors in insect host location. *Behav. Ecol.* **24**, 366-372 (2013).
- 72. Potter, T.I., Stanard, H.J., Greenville, A.C. & Dickman, C.R. Understanding selective predation: are energy and
nutrients important? *PLoS One.* **13**, e0201300 (2018).
- 73. Dussutour, A., & Simpson, S.J. Ant workers die young and colonies collapse when fed a high-protein diet. *Proc.*
*Royal. Soc. B.* **279**, 2402-2408 (2012).
- 74. Jensen, K. et al. Optimal foraging for specific nutrients in predatory beetles. *Proc. Royal. Soc. B.* **279**, 2212-2218
(2012).
- 75. Toft, S. et al. Food and specific macronutrient limitation in an assemblage of predatory beetles. *Oikos.* **128**, 1467-
1477 (2019).
- 76. Felton, G.W. Nutritive quality of plant protein: sources of variation and insect herbivore responses. *Arch. Insect.*
*Biochem. Physiol.* **32**, 107-130 (1996).
- 77. Lee, K.P. The interactive effects of protein quality and macronutrient balance on nutrient balancing in an insect
herbivore. *J. Exp. Biol.* **210**, 3236-244 (2007).
- 78. Simpson, S.J., & Raubenheimer, D. The geometric analysis of nutrient -allelochemical interactions: a case study
using locusts. *Ecology.* **82**, 422-439 (1993).
- 79. Illius, A.W., & Jessop, N.S. (1995). Modeling metabolic costs of allelochemical ingestion by foraging herbivores. *J.*
*Chem. Ecol.* **21**, 693-719 (1995).
- 80. Westoby, M. (1978). What are the biological bases of varied diets? *The American Naturalist.* *112*, 627-631.
- 81. Le Gall, M., & S.T. Behmer, S.T. Effects of protein and carbohydrate on an insect herbivore: the vista from a fitness
landscape. *Integr. Comp. Biol.* **54**, 942-954 (2014).
- 82. Hagele, B.F., & Rowell-Rahier, M. Dietary mixing in three generalist herbivores: nutrient complementation or toxin
dilution? *Oecologia.* **119**, 521-533 (1999).
- 83. Singer, M.S., Bernays, E.A., & Carriere, Y. (2002). The interplay between nutrient balancing and toxin dilution in
foraging by a generalist insect herbivore. *Anim. Behav.* **64**, 629-643 (2002).

- 84. Petit, S., Boursault, A. & Bohan, D.A. Weed seed choice by carabid beetles (Coleoptera: Carabidae): linking field
measurements and laboratory diet assessments. *Eur. J. Entomol.* **111**, 615-620 (2014).
- 85. Gaba, S., Deroulers, P., Bretagnolle, F. & Bretagnolle, V. Lipid content drives seed consumption by ground beetles
(Coleoptera: Carabidae) within the smallest seeds. *Weed. Res.* **59**, 170-179 (2019).
- 86. Frank, S.D., Shrewsbury, P.M. & Denna, R.F. Plant versus prey resources: influence on omnivorous behavior and
herbivore suppression. *Biol. Control.* **57**, 229-235 (2011).
- 87. Blubaugh, C.K., Hagler, J.R., Machtley, S.A., & Kaplan, I. Cover crops increase foraging activity of omnivorous
predators in seed patches and facilitate weed biological control. *Agric. Ecosyst. Environ.* **231**, 264-270 (2016).
- 88. Klimes, P., & Saska, P. Larval and adult seed consumption affected by degree of food specialization in *Amara*
(Coleoptera: Carabidae). *J. Appl. Entomol.* **134**, 659-666 (2010).
- 89. Foffova, H. et al. Which seed properties determine the preferences of carabid beetles seed predators? *Insects.* **11**, 757
(2020)
- 90. Spence, J.R., & Niemela, J.K. Sampling carabid assemblages with pitfall traps: the madness and the method. *Can.*
*Entomol.* **126**, 881-884 (1994).
- 91. Lindroth, C.H. *The Ground Beetles (Carabidae, excluding Cicindelinae) of Canada and Alaska.* (Opusca
Entomology, 1961–1969) Supplement 20, 24, 29, 33, 34, 35. Part I, pages I– XLVIII, 1969. Part II, pages 1–200,
1961. Part III, pages 201–408, 1963. Part IV, pages 409–648, 1966. Part V, pages 649–944, 1968. Part VI, pages
945–1192.
- 92. Honek, A., Martinkova, Z., Saska, P., & Pekar, S. Size and taxonomic constraints determine seed preference of
Carabidae (Coleoptera). *Basic Appl. Ecol.* **8**, 343-353 (2007)
- 93. White, S.S., Renner, K.A., Menalled, F.D., & Landis, D.A. Feeding preferences of weed seed predators and effect
on weed emergence. *Weed. Sci.* **55**, 606-612 (2007).
- 94. Glinwood, R., Ahmed, E., Ovarfordt, E., & Ninkovic, V. Olfactory learning of plant genotypes by a polyphagous
predator. *Oecologia.* **166**, 637-647 (2011).
- 95. Sablon, L., Dickens, J.C., and Haubruge, E.H., & Verhuggen, F.J. Chemical ecology of the Colorado potato beetle,
*Leptinotarsa decemlineata* (Say) (Coleoptera: Chrysomelidae), and potential for alternative control methods. *Insects.*
**4**, 31-54 (2013).
- 96. Zhang, L., Li, H., & Zhang, L. Two olfactory pathways to detect aldehydes on locust mouthpart. *Int. J. Biol. Sci.* **13**,
759-771 (2017).
- 97. Ardenghi, N., Mulch, A., Pross, J., & Niedermeyer, E.M. Leaf wax n-alkane extraction: an optimized procedure.
*Org. Geochem.* **113**, 283-292 (2017).
- 98. Simpson, S.J., & Abisgold, J.D. Compensation by locust for changes in dietary nutrients: behavioral mechanisms.
*Physiol. Entomol.* **10**, 443-452 (1985).
- 99. Lee, K.P., Behmer, S.T., & Simpson, S.J. Nutrient regulation in relation to diet breadth: a comparison of *Heliothis*
sister species and a hybrid. *J. Exp. Biol.* **209**, 2076-2084 (2006).

- 100. Bates, D., Machler, M., Bolker, B., & Walker, S. Fitting linear mixed-effects models using lme4. *J. Stat. Softw.* **67**,
1-48 (2015).
- 101. Behmer, S.T. Insect herbivore nutrient regulation. *Annu. Rev. Entomol.* **54**, 165-187 (2009).
- 102. Nobre, J.S., & Singer, J.D.M. Residual analysis for linear mixed models. *Biom. J.* **49**, 863-875 (2007).
- 103. Schielzeth, H., Dingemanse, N.J., Nakagawa, S., Westneat, D.F., Allogue, H., Teplitsky, C., Reale, D., Dochtermann,
740 N.A., Garamszegi, L.Z., & Araya-Ajoy, Y.G. Robustness of linear mixed-effects models to violations of
741 distributional assumptions. *Methods Ecol. Evol.* **11**, 1141-1152 (2020).
- 104. Takahashi, S., & Gassa, A. Roles of cuticular hydrocarbons in intra- and interspecific recognition behavior of two
Rhinotermitidae species. *J. Chem. Ecol.* **21**, 1837-1845 (1995).

**Table 1.** Seed surface volatile chemicals isolated from the three brassicaceous species used in the experiments showing identity and
average percentage of the detected compounds on measurements of peak areas in GC-MS chromatograms.

Chemical compound	Weed species					
	RT	Formula	CAS #	Brassica napus (n = 5)	Sinapis arvensis (n = 5)	Thlaspi arvense (n = 5)
Nonal	15.97	C ₉ H ₁₈ O	124-19-6	ND	1.97±0.3%	ND
n-Tetradecanoic acid	33	C ₁₄ H ₂₈ O ₂	544-63-8	ND	ND	2.8±0.5%
Hexadecanoic acid ethyl ester	37.23	C ₁₈ H ₃₆ O ₂	626-97-7	ND	< 1%	4.63±1.88% *
E-9-Octadecanoic acid ethyl ester	40.73	C ₂₀ H ₃₈ O ₂	6114-18-7	< 1%	< 1%	43.3±7.76% *
Hexacosane	44.64	C ₂₆ H ₅₄	630-01-3	18±1.45%	11.42±2.87%	< 1% *
Heptacosane	44.87	C ₂₇ H ₅₆	593-49-7	< 1%	31.47±4.44%	< 1% *
Nonacosane	47.43	C ₂₉ H ₆₀	630-03-5	< 1%	17.03±2.17%	< 1% *
15-Nonacosanone	55.42	C ₂₉ H ₅₈ O	2764-73-0	79.98±1.73%	32.46±6.29%	45.66±5.3% *

RT: retention time in minutes; CAS #: Chemical Abstracts Service Registry Number in NIST Mass Spectral Library.

ND: not detected; * Indicates significant quantitative differences between volatiles of weed species as measured by peak area.

**Figure 1.** Seed detection success of sensory-manipulated *Poecilus corvus* (a), *Pterostichus melanarius* (b), and *Amara*
*littoralis* (c) beetles as measured by total number of seeds consumed (mean total seed consumption ± standard error). Seed
choice responses of sensory-manipulated *Poecilus corvus* (d), *Pterostichus melanarius* (e), and *Amara littoralis* (f) beetles
as measured by numbers of seeds consumed from each species (mean number of seeds consumed ± standard error). PC:
positive control; UN: unilateral control, NC: negative control, AP: antennae and palps; AT: antennae; PL: maxillary and
labial palps; MP: maxillary palps; LP: labial palps; E: eyes; +/+ : positive control (intact insects); +/-: unilateral control; -/-
: negative control.

**Figure 2.** Seed volatile chemical compounds detected in surface seed extracts of *Brassica napus* (a), *Sinapis arvensis* (b),
and *Thlaspi arvense* (c) measured as total ion content TIC (ion abundance) in *mV*. Numbers represent compounds: (1)
Nonal, (2) n-Tetradecanoic acid, (3) Hexadecanoic acid ethyl ester, (4) E-9-Octadecanoic acid ethyl ester, (5) Hexacosane,
(6) Hepatocosane, (7) Nonacosane, (8) 15-Nonacosanone.

**Figure 3.** Feeding responses (mean food consumption \pm standard error) of three carabid species offered protein pellets
coated with hexane against protein pellets coated with *B. napus* seed surface chemicals (a), and protein pellets coated with
*B. napus* surface chemicals against protein pellets coated with surface chemicals of *T. arvense* seeds (b) in two-choice
feeding bioassays. Feeding responses (mean food consumption \pm standard error) of three carabid species offered protein
pellets coated with *B. napus* surface chemicals against protein pellets coated with surface chemicals of *S. arvensis* and *T.*
*arvense* seeds in three-choice feeding bioassays (c). Feeding responses (mean number of seeds consumed \pm standard error)
of three carabid species offered seeds of *B. napus* coated with the surface chemicals of *T. arvense* against uncoated intact
*B. napus* seeds (d), and seeds of *T. arvense* coated with the surface chemicals of *B. napus* seeds against uncoated *T. arvense*
(e) in two-choice feeding bioassays.

**Figure 4.** Protein (P) and lipid (L) ingestion rates and bivariate intake points showing estimates nutrient ingestion rates of
*Poecilus corvus* (a and b respectively), *Pterostichus melanarius* (c and d respectively), and *Harpalus amputatus* (e and f
respectively) beetles under balanced and unbalanced protein-to-lipid dietary conditions. aa: lipid-biased conditions; bb:
balanced conditions; cc: protein-biased conditions; error bars: mean nutritional intake \pm standard error.

**Figure 5.** Protein (P) and lipid (L) ingestion rates and bivariate intake points showing estimates nutrient ingestion rates
*Poecilus corvus* (a and b respectively), *Pterostichus melanarius* (c and d respectively), and *Harpalus amputatus* (e and f
respectively) beetles under balanced and unbalanced protein-to-lipid dietary conditions and low-quality protein. aa: lipid-
biased conditions; bb: balanced conditions; cc: protein-biased conditions; error bars: mean nutritional intake \pm standard
error.

**Figure 6.** Protein (P) and lipid (L) ingestion rates and bivariate intake points showing estimates nutrient ingestion rates
*Poecilus corvus* (a and b respectively), *Pterostichus melanarius* (c and d respectively), and *Harpalus amputatus* (e and f
respectively) beetles under balanced and unbalanced protein-to-lipid dietary conditions with protein quality high and ally
isothiocyanate added at 0.5% (*v/v*). aa: lipid-biased conditions; bb: balanced conditions; cc: protein-biased conditions; error
bars: mean nutritional intake \pm standard error.

**Supplementary Information**

*Dynamic headspace sampling of seed volatiles*

Dynamic headspace sampling was used to collect samples of weed seed volatile organic compounds VOCs
from the three brassicaceous weed species mentioned above. The dynamic headspace sampling was carried out
using a Sigma Air Delivery System (Sigma Scientific, USA). The sampling procedure was initiated by placing a
mass of 500 mg of imbibed seeds on a 2 × 4 cm clean filter paper into a clean and sterilized glass odor collection
chamber (Sigma Scientific, USA). At one end, the chamber was connected to source of clean and filtered air. On
the other end, a VOC collection trap was attached to the chamber. The VOC traps were built by using 3.5'' clean
and sterilized Pasteur glass pipettes 5 mm internal diameter (Sigma Scientific, USA). In each glass pipette, Porpak
Q (150 mg, 80 – 100 µm) was positioned between 3 glass wool plugs (100 + 50 mg of Porpak Q, respectively).
All volatile traps were covered with a layer of aluminum foil since Porpak Q is reactive to light. Volatile sampling
was carried out by pushing clean and filtered air into the glass chamber containing the seed mass, and then through
the VOC trap. Volatile sampling via the air entrainment system for each seed mass (replicate) was maintained for
48 hr. Five independent collections were carried out per each weed species, each representing a replicate. Also,
a blank sample (filter paper only) was used as a negative control for each round of VOC sampling. Between
sampling sessions, all glassware was washed with three rinses of *n*-hexanes, three rinses of ethanol, and three
rinses acetone, then baked in a dry oven at 130 °C overnight. At the end of the air entrainment sessions, the
collected volatiles were eluted (desorbed) from the Porapak Q traps with 3 ml of *n*-hexanes (HPLC-grade)
containing 250 ng of pentadecane as an internal standard. Following elution, extracts were concentrated down to
200 µl under a gentle stream of liquid nitrogen and then stored at -80 °C until GC-MS analysis.

*Static headspace sampling of seed volatiles using Solid-Phase Microextraction (S-SPME) fibers*

Seed volatile chemicals were sampled in this experiment by placing a 500 mg mass of imbibed weed seeds in
a clean and sterile 5 ml crimp-top glass tube. An SPME fiber coated with polydimethylsiloxane (Supelco, Sigma
Aldrich, Canada) was then inserted through the tube into the vial and positioned above the seed mass without
contacting it. The preparation was then incubated at 21±1 °C in a growth chamber for 24 hours. Five independent
preparations were per each weed species each representing a replicate. The same steps were repeated, but without
placing seed masses in the glass tube, and those blank preparations served as a negative control. After incubation,

any chemicals trapped on the fiber were extracted by thermo-desorption at 250 °C for 5 minutes, then injected
into the GC column for thermal fractionation. GC conditions and thermal fractionation were as described in GC-
MS analysis section.

*Chemical coating of protein pellets with seed surface extracts*

Chemical coating was performed by soaking the pellets in 2 ml of a specific concentrated seed extract for 30
832 min. Pellets after soaking were placed on clean filter paper and left for 10 min for the hexanes to evaporate.
Coated protein pellets were then used in three sets of multiple-choice feeding experiments. In the first experiment,
pellets coated with seed extracts of *B. napus* were offered to *P. melanarius*, *P. corvus*, or *H. amputatus* carabids
against pellets coated with *n*-hexanes only in two-choice feeding bioassays. In the second experiment, pellets
coated with surface extracts of *B. napus* seeds were offered to *P. melanarius*, *P. corvus*, or *H. amputatus* carabids
against pellets coated with surface extracts of *T. arvense* seeds in two-choice feeding bioassays. In the third
experiment, three patches of protein pellets were used: one coated with surface extracts of *B. napus* seeds, the
other coated with surface extracts of *S. arvensis* seeds, and the third coated with surface extracts of *T. arvense*
seeds. Pellets in this experiment were offered to *P. corvus*, *H. amputatus*, and *A. littoralis* carabids in three-choice
feeding experiments. Carabids were released into the feeding arenas following 72 hours of starvation, and were
allowed to feed for 5 consecutive days. Feeding experiments were replicated 10 times across treatments and
carabid species.

*Manipulation the surface chemistry of seed species via chemical coating techniques*

Chemical coating techniques were adopted to manipulate the surface chemistry of seed species themselves.
The aim of this experiment was to test if seed surface chemistry, and therefore their preferability to carabid
predators, could be changed by the chemical coating procedures. Treatments here represented coating seeds of *B.*
*napus* (highly preferable to carabids) with surface extracts of *T. arvense* seeds (weakly preferable to carabids).
The chemical coating was carried out by soaking the seeds in 2 ml of concentrated seed extract suspended in
Triton X-100 (2% v/v) with for 30 min.¹⁰⁴ Coated and uncoated *B. napus* seeds (25 seeds of each) were then
placed in plasticine trays and offered in two-choice Petri-dish feeding bioassays to *P. corvus*, *H. amputatus*, or
*A. littoralis* carabids following after 72 hours of starvation. Petri dishes were kept in a growth chamber at 21±1°C
and 16:8 L:D photoperiod. Feeding trails were replicated 10 times for each species, and insects were allowed to
feed for 5 consecutive days. Carabids were removed at the end of the experiment and seed consumption rates

were recorded. The exact same steps were repeated to coat seeds of *T. arvensis* with surface extracts of *B. napus*
seeds and offer them to carabids in two-choice feeding arena.

*Two-choice feeding bioassays using synthetic diets*

Prior to feeding experiments, food blocks were weighed to the nearest 0.1 mg of fresh mass, and then each
food block was placed into one side of a Petri dish. After that, a single adult predatory beetle was released into
the Petri dish following 73 hr of starvation. Petri dishes were then placed into a growth chamber at 21 ± 1 °C and
16:8 L:H photoperiod. Beetles were left to feed on the synthetic food for 24 hours. After 24 hours, food blocks
were replaced with new ones, and the remaining food (i.e. removed blocks) were dried to a constant mass in a
desiccation oven (at 50°C for 24 hours). After drying, food remnants were weighed to the nearest 0.1 mg of dry
mass. This daily protocol was repeated for five consecutive days of feeding. Throughout the experiments, each
beetle was used only once, and treatments were replicated 14 times for *P. corvus*, and *P. melanarius*, and 12 times
for *H. amputatus*. Carabids were removed at the end of the experiment and food consumption was calculated.
The fresh mass (mg) for every carabid used in the experiments were recorded by weighing the carabid to the
nearest 0.1 mg after starvation and prior to its release in the Petri dishes.

*Measurements of synthetic dietary intake*

The dietary intake measurements were based on dry mass values of the diet blocks. This required estimating
the relationship between fresh mass and dry mass for the food blocks. For this purpose, the ~~exact~~ same
experimental protocols described above were repeated but with no insects being released into the petri dishes.
These data were used to establish the relationship between fresh and dry masses of the food blocks via regression
analysis ($n = 20$ blocks per each P:L diet). Regression equations were used to convert fresh mass values to dry
mass values. Daily food consumption was then calculated as the difference between initial and final dry masses
of food blocks offered to the beetles in each Petri dish (mg food dry mass). The cumulative food intake for each
beetle was calculated by adding up the values of daily food intake. Finally, the amounts of protein and lipid (mg
dry mass) ingested by the beetles over five days were calculated by multiplying total food consumption by the
corresponding P:L ratio of the diet block.

Commented [A13]: L:D photoperiod

**Modeling the interaction between carabid predators and weeds seeds in the model system under study**

 Mixed effects models using the function “lmer” were used for modeling the interaction between carabid species
 and weed seeds used in the study. The experimental design had spatial nestedness (three seed species nestled in
 each Petri dish); therefore, replicate was used as a random blocking factor in the model to account for the error
 structure in the design.

 **Table 2.** Mixed effects analysis (P-values) for measured seed feeding responses of the full dataset of the three tested carabid species as
 affected by sensory manipulation treatments, weed species, insect species, and insect sex and their interactions.

Statistical term	ndf	ddf	F-value	P-value
Weed species	2	1552	10.99	P < 0.0001
Sensory treatment	8	1552	55.73	P < 0.0001
Insect species	2	1552	8.6	P < 0.0001
Insect sex	1	1552	0.01	P = 0.9
Weed Species × Sensory treatment	16	1552	2.27	P = 0.002
Weed Species × Insect species	2	1552	74.7	P < 0.0001
Sensory treatment × Insect species	16	1552	2.22	P = 0.003
Sensory treatment × Insect sex	8	1554	0.94	P = 0.47
Insect species × Insect sex	2	1543	0.16	P = 0.85
Weed species × Sensory treatment × Insect species	32	1531	8.77	P < 0.0001
Weed species × Sensory treatment × Insect sex	16	1552	0.8	P = 0.68
Weed species × Insect species × Insect sex	4	1552	0.86	P = 0.48
Sensory treatment × Insect species × Insect sex	16	1542	1.41	P = 0.12
Weed Species × Sensory Manipulation × Insect Species × Insect sex	32	1554	1.09	P = 0.32

*ndf*: numerator degrees of freedom; *ddf*: denominator degrees of freedom

 *Modeling the nutritional ecology of carabid predators in the model system under study*

 The model was created via mixed-effects modeling using the function “lmer” in R Package. Total amounts of
 protein and lipid consumed were used as the response variable measuring feeding responses of carabid predators
 under the different synthetic nutritional conditions created in the experiment. Replicate was used as a random
 blocking factor in the model to account the spatial nestedness of the design (two diet locks nestled in each Petri
 dish). Protein-to-lipid ratios in the food blocks was also included in the random term of the model.

**Table 3.** Mixed effects analysis (P-values) for measured feeding responses of three carabid species on different synthetic diets of a
 moderate allyl isothiocyanate dosage as affected by protein-lipid P:L ratio, insect sex, and their interactions.

Statistical term	ndf	ddf	F-value	P-value
Macronutrient	2	1049	10.46	P < 0.001
Nutritional landscape	2	1047	16.9	P < 0.001
Toxicity	1	1050	24.66	P < 0.001
Diet quality	2	1042	36.43	P < 0.001
Insect species	2	1047	18.73	P < 0.001
Insect sex	1	1050	1.16	P = 0.24
Macronutrient × Nutritional landscape	2	1045	41.2	P = 0.001
Macronutrient × Toxicity	1	1045	10.15	P < 0.001
Nutritional landscape × Toxicity	2	1045	10.43	P < 0.001
Macronutrient × Diet quality	4	1041	3.56	P = 0.006
Macronutrient × Insect species	3	1045	1.45	P = 0.22
Nutritional landscape × Diet quality	4	1041	8.45	P < 0.001
Nutritional landscape × Insect species	4	1048	4.03	P = 0.002
Toxicity × Insect species	2	1052	0.55	P = 0.56
Diet quality × Insect species	4	1045	24.39	P < 0.001
Macronutrient × Insect sex	2	1041	17.95	P = 0.84
Nutritional landscape × Insect sex	2	1058	1.24	P = 0.28
Toxicity × Insect sex	1	1040	2.17	P = 0.14
Insect species × Insect sex	2	1040	1.04	P = 0.35
Macronutrient × Nutritional landscape × Toxicity	2	1045	2.28	P = 0.1
Macronutrient × Nutritional landscape × Insect species	4	1045	2.64	P = 0.03
Macronutrient × Nutritional landscape × Diet quality	4	1041	15.88	P < 0.001
Macronutrient × Toxicity × Insect species	2	1045	1.31	P = 0.26
Nutritional landscape × Toxicity × Insect species	4	1041	4.31	P = 0.001
Macronutrient × Toxicity × Insect sex	2	1041	3.42	P = 0.03
Macronutrient × Diet quality × Insect species	4	1041	0.69	P = 0.59
Nutritional landscape × Toxicity × Insect sex	1	1050	0.57	P = 0.56
Nutritional landscape × Diet quality × Insect species	8	1041	1.96	P = 0.47
Macronutrient × Insect species × Insect sex	2	1045	0.26	P = 0.76
Macronutrient × Nutritional landscape × Diet quality × Insect species	8	1041	2.18	P = 0.026
Macronutrient × Nutritional landscape × Toxicity × Insect species	4	1055	1.71	P = 0.14
Nutritional landscape × Insect species × Toxicity × Insect sex	2	1045	0.45	P = 0.63
Nutritional landscape × Insect species × Insect species × Insect sex	4	1045	0.35	P = 0.84
Macronutrient × Toxicity × Insect species × Insect sex	2	1045	0.22	P = 0.79
Nutritional landscape × Toxicity × Insect species × Insect sex	4	1055	0.86	P = 0.48

913

Referee expertise:

Referee #1: Carabid beetles

Referee #2: Evolutionary biology, weed seed research

Referee #3: Molecular insect physiology

Reviewers' comments:

Reviewer #1 (Remarks to the Author):

I enjoyed reading this work. This work helps me to answer some of the questions that I have on the topic of carabid preferences and choices. In it originally and shows how the chemical compounds of seeds attract their predator. This topic is really important because there is not enough information about it. It can help us understand and also predict the ecosystem service called seed predation (which can be nowadays very important with the European program of Green Deal).

I do not understand properly some things or I have to disagree. Mostly I wrote them to the attached word document to make it easier to find (hope that it will be more practical for you too).

RE: Changes have been made following the suggestions and comments in the word document, please see ((LL92-93; 185-192; 350-353)).

In the introduction and also in the discussion I miss any note that it is not “just” chemical compounds that drive the preference but also some physical properties (such as size, mass,

shape etc.) – these properties of the seeds were detected as a driver many times (for example work of Saska et al. 2019 or Honek et al. 2003).

RE: In this study, we focus only on the sensory and chemical aspects of seed detection and selection. We added a few lines in the Introduction to clarify that chemically-driven seed choices in carabids hold only when other seed physical traits do not vary widely among the seed species accessible to the carabid predator. ((LL 97-103)). Also, we added a paragraph in the Discussion to address the potential impact of seed physical traits, along with other biotic and abiotic factors in the environment, on seed selection decisions in carabids. This addition is likely to clarify how volatile-guided seed selection can, among other things, fit into the general picture of seed selection decisions. ((LL 283-302))

In M&M:

Why did you use just some species and not all of them in all experiments? I am sure that for example, the full data set for *Amara* would be nice and it could show differences between granivorous (*Amara*) and predatory carabids (*Pterostichus*). It would also improve the paper.

RE: A detailed paragraph has been added to the Methods section clarifying the reason why not the same species were used across all experiments. We argue that some ecological similarities between certain carabid species in the current study allowed for replacing one by the other when species of interest was not abundant in the catches. Thus, the species inconsistency in some of the experiments is unlikely to undermine the validity of our ecological inferences since all species tested are as omnivorous in ecological terms, regardless of their dietary breadth. The impact of dietary breadth on seed preference was not investigated in this study, and its effects remain to be explored in future studies. Please see ((LL 332-342)) for a more detailed explanation.

How do you know that the volatiles was just the volatiles from the seeds and not from some microorganisms on the surface of the seeds?

RE: In the new version of the manuscript, we added an argument in the Discussion section to clarifying why we think it is highly unlikely that the seed volatiles isolated and identified in the current study had originated from the seed microbiome. We clarify that long chain alkanes, ketones, and esters are hallmarks of plant surface hydrocarbons, which are species specific and not highly volatile. By contrast, microbial volatiles are often of less than C₁₅, highly volatiles, and of low molecular mass which was not the case for the major seed

volatiles isolated and identified in this study. Please see ((LL 262-269)) for a more detailed argument.

How did you collect and clean the seeds? Did you use gloves? Did you dry the seeds before storage? Could you please provide more information about this time? Was the experiment made in the same year as the seeds were collected?

RE: We added more details to the Methods to clarify aspects of seed collection, handling, and storage. For example, seeds were used one year after collection as fresh seeds were not available for experimentation at the same year of the study. Seeds were thus stored (at 5 c) after collection, and were not dried prior to storage. ((LL 315-318))

Reviewer #2 (Remarks to the Author):

This paper aims to study the sensory biology of several species of granivorous or omnivorous ground beetles, which frequent cultivated fields. The authors therefore tackle key processes that can lead to the regulation of weed species in crops by manipulating ecological processes (here granivory). I salute the enormous amount of information and work carried out through this article as well as the originality of the data presented in it. I therefore think that all these data provide original and important knowledge both to dissect the food decision strategies of these insects but also to define nature-based control strategies of weeds.

This being clearly stated, I believe that the paper is not publishable in the form currently presented and that it requires a lot of rewriting. The first reason has to do with the astronomical amount of experiences and results that are presented. The second reason is linked to the complexity of reading the manuscript will develop these two points.

Regarding the first point, I think there are too many different experiences, each one being complex, that are presented. The reader, faced with this very large quantity of experiences which have been conducted and which are presented, very quickly gets lost in the text. This article deserves to be split into two different papers, the first focusing on highlighting, on several species of ground beetles, the relative role of different sensory organs in the detection of different species of seeds, the second being devoted to the different constituents of seeds and their role in the choice strategies. I have the feeling, reading this paper, that these results present the whole of a doctoral work. If the authors wish to present all the results in the same paper, I think that it will be necessary to clearly define subcategories of experiments and

indicate, at the end of the introduction, in a dedicated paragraph, where the different experiments are positioned and the questions they wish to answer.

RE: We agree and so following the Reviewer and Editor recommendations, we have removed the nutritional ecology work from the new version of the ms, which will be prepared as a separate ms.

I therefore think that the final paragraph of the introduction (L90-99) should be developed in the logic of the paper and supported by bibliographical references). The authors indeed state a certain number of facts (e.g. ... omnivorous carabids use olfactory perception ... (L. 90) ... seed choice is driven by the perception of long chain volatiles ... (L91). .. seeds volatiles seem to encode information about the lipid content (L. 93) ...) for which no references are given.

RE: We made all the recommended changes in the revised version of the manuscript, and references have been added to support our claims were relevant. ((LL87-103))

Regarding the second point, I had a lot of trouble going into each experience that was done with regard to their presentation because to clearly understand the experiments that were carried out and to interpret the figures on this basis, I had to go back and forth complex between the results, the material and methods and the supplementary information. Furthermore, the presentation logic in the figures is complex. With regard to the first experience (manipulation of the sensory organs), the name of the treatments and what they correspond to, is not logically and clearly formulated. For example, does the "eyes" treatment correspond to "all sensory organs except the eyes are neutralized" or "" the eyes are neutralized "? Does the " maxillary palps "treatment correspond to" all sensory organs other than maxillary palps are neutralized "or" "maxillary palps are neutralized"? In general, the description of sensory treatments is poorly written. It is necessary to clearly indicate which organs have been manipulated (for example the authors speak of gustatory organs in the text without indicating that there are two visibly, the maxillary and the labial palps) and respect a logic of identification of treatment which will be found in the figures (eg the "Palps" treatment correspond to individuals for whom both the maxillary and labial palps have been neutralized).

RE: We have rewritten the sensory manipulations section in a way that better describes the methodology and the sensory manipulations they brought about in the treated carabids. We also added Table 2 to summarize the treatments and simplify comparisons between treatments. Please see ((LL 364-373)) for more details.

The different species used in the different experiments are not the same which can potentially be a problem. The authors should justify this point. For example, why the species *Harpalus* sp. is not used in the experiment shown in Figure 1? Why in the series of experiments with coated pellets or seeds (L. 153-169) it is not the same species which are always used. This could prevent drawing general conclusions.

RE: A detailed paragraph has been added to the Methods section clarifying the reason why not the same species were used across all experiments. We argue that some ecological similarities between certain carabid species in the current study allowed for replacing one by the other when species of interest was not abundant in the catches. Thus, the species inconsistency in some of the experiments is unlikely to undermine the validity of our ecological inferences since all species tested are as omnivorous in ecological terms, regardless of their dietary breadth. The impact of dietary breadth on seed preference was not investigated in this study, and its effects remain to be explored in future studies. Please see ((LL 333-343)) for a more detailed explanation

Regarding this same series of experiments, a general table presenting all the treatments that have been carried out would be of great use to help the reader understand everything that has been done and guide him in reading the results and figures.

RE: We have noticed that experimentation with protein pellets took much space and cause a lot of confusion in the initial manuscript. Plus, results produced by experimentation with chemically-coated seeds were quite similar to experiments with protein pellets. For these two reasons, protein pellets experiment was removed from the revised manuscript to simplify the structure and presentation of the behavioral study. The simpler version of this section negated the need for a table to summarize the treatments and their results as recommended.

This last remark is even more important with regard to the series of experiments which were carried out to test the role of the protein quality and the protein-lipid ratio in the seeds. Here again, it is necessary to clearly qualify the treatments and indicate, for example, that the "protein-biased" treatment corresponds to 21:21 versus 35: 7 in the choice experiments carried out. In these experiments, the presentation of geometric framework bivariate analysis is too insufficient to be able to clearly read what the graphs present.

RE: The nutritional ecology section has been removed from the manuscript and so this point will no longer be addressed by this manuscript.

To conclude on something positive, and I think there is plenty of material for this, this paper presents a set of very important experiences for understanding decision-making strategies in insect species that have been identified as key potentials regulating actors of weeds in agricultural fields. But I think a lot of work is needed to make the manuscript readable for the reader.

Reviewer #3 (Remarks to the Author):

Ms title: The interplay between seed volatiles and lipids drives seed choice in ground beetles
Ms #: COMMSBIO-21-3313

This paper demonstrates that seed preference of omnivorous ground (carabid) beetles is linked to the long chain volatile chemicals derived from the epicuticular lipids on the seed coat surface, leading to assessment of the fatty acid content of the seed and accurate seed choice. Seeds of three brassicaceous species (*Brassica napus*, *Sinapis arvensis*, *Thlaspi arvense*) and adults of four carabid species *Poecilus corvus*, *Harpalus amputatus*, *Pterostichus melanarius*, and *Amara littoralis* were used in the study. Additionally, accurate seed choice was demonstrated to occur through the antennae and palps. Overall, I found the paper interesting; however, I will have various points to be addressed/considered.

1. Title: I found the title inappropriate for a research paper as it sounds like a title of a review paper. I recommend authors to come up with a more specific & conclusive & definitive title.

RE: The title has been changed to a more specific and definitive one.

2. Abstract: Abstract includes general statements and does not include the details of the study. So rather than mentioning common statements, authors should highlight their specific data as the abstract should stand by itself. The tested species and seed species should be also mentioned.

RE: Following this recommendation, the Abstract has been rewritten and slanted towards a more specific account. Experimental species have also been mentioned in the new abstract.

Line 18: “....numerous species are predators of various pests and weed seeds”. It is not accurate to make a definition as “predator of weed seeds”. I recommend smthng like “.....numerous species are predators of various pests and feed on weed seeds at the same time”.

RE: Corrected. ((LL 16-18))

3. Introduction:

Line 31: Pls replace parasite with parasitoid.

RE: corrected.

4. Results:

Line 188: “.....different landscapes in both species as the lipid intake...” Which both species?

RE: The nutritional ecology section has been removed from the manuscript and so this point will no longer be addressed by this manuscript.

Lines 190-191: “Together, these observations suggest that this carabid species is likely lipid-limited and modulate their protein intake to address this limitation.” How can protein intake be modulated to compensate lipid limitation? Pls explain or revise or omit.

RE: The nutritional ecology section has been removed from the manuscript and so this point will no longer be addressed by this manuscript.

5. Discussion:

Lines 253-254: “Maxillary and labial palps appear to carry significantly fewer of those chemoreceptors, as predators carrying one pair of either palps failed to make accurate seed choices”. Is this interpretation correct? If the palps possess significantly fewer chemoreceptors, how come predators with one pair of either palps can fail to make accurate seed choice? Pls explain or revise or omit.

RE: Changes have been made to this section in the Discussion to clarify in detail why we think carabids need enough olfactory receptors to gather the sensory information necessary for accurate seed selection. Please see ((LL 204-218))

Lines 256-257: "Olfactory receptors can also be found on maxillary and labial palps, but their abundance on these appendages is usually rather low". This was already mentioned in Lines 253-254. Please avoid of repetitions.

RE: Corrected.

Lines 259-261: "Nonetheless, some authors suggest that sensory information perceived through olfactory receptors is usually more accurate and specific than information perceived via gustatory receptors 55". You say some authors but cite a single paper.

RE: Corrected. ((LL 212-215))

Lines 268-269: "The fatty-acid-derived seed cues are located on seed surfaces and comprise three main groups of long-chain aliphatic lipids: alkanes, esters, ketones". Reference needed.

RE: Corrected.

Lines 292-294: "Based on our data we suggest that seed surface lipids provide the "kairomonal" signals necessary for interactions between carabid predators and seeds of weed species to initiate and unfold 60". You make a suggestion and cite a paper. Is this your suggestion or Ref 60s'? Please revise.

RE: Corrected. We use this reference to support the logic of our suggestion that seed surface hydrocarbons can act as the kairomonal signals in seed predation systems. ((LL247-255))

Lines 335-342: "The theme/idea here was already mentioned in lines 309-319. So please avoid of repetition.

RE: This section has been omitted.

6. Material and Methods:

- Section on "Elucidating the sensory mechanisms of seed detection and discrimination in carabid": How many beetles were used per tray?

RE: We state in the Methods that a single carabid was released into each Petri dish. ((LL 381-382))

Line 405: "...ablating one or more of the sensory appendages with precision tweezer".
The appendages should be specified.

RE: This section has been re-written toward a more specific account, with a more detailed description of the sensory treatments and how we were able to produce different groups of carabid beetles, each lacking the ability to perceive sensory information of a specific nature (i.e. vision, olfaction, or gustation). ((LL 364-373))

- Section on "Isolation, identification, and behavioral testing of seed cues that mediate seed perception in carabids": Lines 441-442: "The impact on seed foraging behaviors of seed volatile cues identified above was tested via coating protein pellets with surface extracts of different seed species.". Is there a reference for this method or this is a novel methodology introduced here in this paper?

RE: This section has been re-written to better describe and clarify the different aspects of chemical coating of seed species. Chemical coating of seed species was a novel method developed for the purposes of this study. ((LL 413-427))

Line 442: "The use of protein pellets (100% shrimp protein and no fat)...". How was a protein pellet without any fat was prepared from Shrimp or was it purchased?

RE: This section has been omitted.

Lines 443-445: "However, protein pellets offered a simple and homogenous (physically and chemically) food source to measure carabid feeding responses to seed surface extracts under controlled conditions." Reference needed.

RE: This section has been omitted.

REVIEWERS' COMMENTS:

Reviewer #1 (Remarks to the Author):

Dear authors,
this is the second review of this paper.

I have to say that there is an improvement in the manuscript and that the manuscript is easier to read. But still, I have some questions, suggestions and confusion.

I miss more information - how carabid beetles can find the seeds in the introduction. There is no information on why did you decide to manipulate antennae, palps or eyes. I think that such information should be in the introduction. I realised that you mentioned it in the discussion. But I would prefer them in the introduction.

In the discussion, I would like to see a better discussion about results. I missed the discussion about seed predators, which are more specialised (*Amara* and *Harpalus*) versus occasional seed predators.

In M&M - L382 - feeding arena - Was a Petri dish closed or open? It may play a role in a mixture of volatile compounds.

One more question: How many carabids died during the experiments.

I still think that the easiest explanation for why *Amara* species preferred to consume *T. arvense* is that the size of the seed is different from the other seeds. This explanation can work another way - that bigger carabids consumed the big seeds the most. This idea came out of the works of Alois Honěk.

I have more comments. I put them into the word document - to make them easier to find. The document is attached.

Reviewer #2 (Remarks to the Author):

After rereading this article, which corresponds to a revision, I am fully satisfied with the modifications made by the authors. The manuscript has gained in clarity and the fact of having chosen to delete part of the experiments presented makes it possible to lighten the text. I therefore recommend this article for publication.

Reviewer #3 (Remarks to the Author):

Removal of the sections on nutritional ecology work and the experimentation with protein pellets from the behavioral studies made sense and I found this new revised & reduced version satisfactory. I have no new suggestion.

Seed choice in ground beetles is driven by surface-derived hydrocarbons

Khaldoun A. Ali^{1*} Boyd A. Mori² Sean M. Prager¹ and Christian J. Willenborg¹

¹ Plant Sciences Department, College of Agriculture and Bioresources, University of Saskatchewan, Saskatoon, Saskatchewan, Canada

² Department of Agricultural, Food and Nutritional Science, University of Alberta, Edmonton, Alberta, Canada

* Corresponding author: Department of Plant Sciences, University of Saskatchewan, Saskatoon, SK, Canada S7N 5A8 (chris.willenborg@usask.ca)

Keywords: Carabidae, agroecology, weed biological control, ecosystem service, seed preference, seed volatiles, seed nutrients, seed defensive chemicals.

Abstract

Ground-~~(carabid)~~ beetles (Coleoptera: Carabidae) are among the most prevalent biological agents in temperate agroecosystems. Numerous species function as omnivorous predators, feeding on both pests and weed seeds, yet the sensory ecology of seed perception in omnivorous carabids remains poorly understood, however. Here, we explore the sensory mechanisms of seed detection and discrimination in four species of omnivorous carabids: *Poecilus corvus*, *Pterostichus melanarius*, *Harpalus amputatus*, and *Amara littoralis* (~~Coleoptera: Carabidae~~). Sensory manipulations and multiple-choice seed feeding bioassays showed olfactory perception as the primary mechanism used by omnivorous carabids to detect and distinguish among seeds of *Brassica napus* L., *Sinapis arvensis* L., and *Thlaspi arvense* L. (Brassicaceae). Seed preferences differed among carabid species tested, but the choice of desirable seed species was generally guided by long chain hydrocarbons derived from the seed coat surface. Previous literature suggests that seed surface hydrocarbons encode chemical information about the fatty acid composition of seed species. Carabids are, therefore, predicted to exploit seed surface hydrocarbons to identify the suitable seed species based on lipid content.

Introduction

[revised manuscript text omitted]

Northern Great Plains region of North America.²⁷ Previous work has shown that these seed species varied widely
in their palatability to carabid seed predators.^{27,32} Accordingly, seeds of canola (*B. napus*) were used as a highly
preferable seed species, whereas seeds of wild mustard (*S. arvensis*), and field pennycress (*T. arvense*)
represented moderately and weakly acceptable seed species, respectively. Seed masses of the three weed species
were obtained from stored samples collected in summers of 2016-17. Seeds were collected from different field
sites at the Kernen Crop Research Farm near Saskatoon, SK, Canada (52°09'10.3" N 106°32'41.5" W), and then
stored at 5°C to be used the next year after collection. Seeds were not dried before storage, and contact between
the skin and seed surfaces was avoided by wearing gloves and using sterile tweezers during seed handling.

Commented [A5]: I would suggest to delate this. It is confusing because B. napus is a crop as well. I understand the fact that it can be weed in the next years.

Commented [A6]: I miss citation here.

Commented [A7]: How much?

*Carabids*

Adults of the omnivorous carabid species *Poecilus corvus*, *Harpalus amputatus*, *Pterostichus melanarius*, and
*Amara littoralis* were used in this study as all are known to consume weed seeds. Live adults were collected from
different field sites at the Kernen Crop Research Farm in the summers of 2018-19 via dry pitfall trapping. Field
sites chosen for carabid trapping were seeded with canola, pulse, or cereal crops. Pitfall traps consisted of two
plastic 0.5 L cups (10 cm height × 8 cm diameter). One cup acted as a sleeve and with its lip kept flush with the
soil surface, and the other cup (the actual trap) was inserted into it.⁸³ Pitfall traps were enclosed into cages of fine
wire mesh ($\sigma = 1.1$ cm) to prevent vertebrates from entering the traps and ravaging the catches. Traps were
emptied every three days and the collected insects were placed into plastic boxes (40 cm × 25 cm, 25 cm depth)
lined with plant material and moist filter paper. Boxes were transported to the laboratory for identification and
experimentation. Carabid species identity and sex of the experimental carabids were determined using keys in
Lindroth (1961-1969).⁸⁴

Commented [A8]: Please use word "legume"

The abundance of *P. melanarous*, *P. corvus*, *H. amputatus* and *A. littoralis* in the catches fluctuated among the
seasons. Therefore, there was not always adequate numbers of *P. melanarous*, *P. corvus*, *H. amputatus* or *A.*
*littoralis* for the behavioral experiments. All of the carabid species used in this study feature omnivorous feeding
habits, but *H. amputatus* and *A. littoralis* are more specialized towards seed feeding, which makes them more
ecologically similar compared to the other species in the experiment.²⁴ Similarly, *P. melanarous*, *P. corvus* are
both omnivorous but not specialized towards seed feeding like the other two species in the experiments.²⁴ These
ecological similarities between carabid species in the current study allowed for replacing one with the other if a
particular species was not abundant in the catches. Thus, given the omnivorous nature of the included carabid
species, the variability of species in some of the experiments is unlikely to undermine the validity of our

Commented [A9]: I totally disagree – You can not replace them so easily. In the literature, there are not many information about *A. littoralis* preferences and diet. And there are already papers which show that each species has different preferention.

ecological inferences since all species tested are as omnivorous in ecological terms, regardless of their dietary

[revised manuscript text omitted]

Commented [A10]: Please add citation – because in your paper Amara consumed the *T. arvense* the most.

Mixed effects models using the function “lmer” were used to compare the number of seeds consumed by each
carabid beetle from each seed species under sensory treatments.⁹³ Data were analyzed and presented for each
species separately as significant differences between carabid species were detected in the full data set (all three
carabid species together). For each predatory species, the analysis was initiated by fitting a maximal model to the
data including weed species, sensory treatment, and insect sex and their possible interactions as the main effects.
Replicate was used as a random blocking factor in the model to account for spatial structure in the experiment
(three seed species placed into each Petri dish).

Mixed effects models were also used to compare peak areas (GC-MS chromatograms) of the chemical
compounds identified for seed species. A maximal model including weed species, identity of chemical compound,
and their possible interactions as the main effects was fitted to the data. Replicate was used as a random blocking
factor as compounds were nested in weed species. Similar mixed effects modeling steps were used to compare
the number of coated and uncoated seeds consumed by carabids in the two-choice seed coating experiments. A
maximal model was fitted to the data including seed treatment, insect species, and their possible interactions as
the main effects. Replicate was used as a random blocking factor as above (two patches of seeds nested in each
Petri dish).

Model validity was checked by examining the distribution of model residuals throughout the mixed modeling
steps described above.⁹⁴ The R package “LmerTest” was used to perform post-hoc comparisons on the final
models.⁹⁵

**Acknowledgements**

We would like to thank Agriculture and Agri-Food Canada (AAFC) and the Saskatchewan Structural Sciences Center
(SSSC) for providing excellent technical support. Funding was provided by a Discovery Grant awarded to CJW by the
Natural Sciences and Engineering Research Council of Canada. Financial support was provided to BAM by a Natural
Sciences and Engineering Research Council of Canada Industrial Research Chair (Grant 545088) and partner organizations
Alberta Wheat Commission, Alberta Barley Commission, Alberta Canola Producers Commission, and Alberta Pulse
Growers Commission during the preparation of the manuscript.

**Author Contributions**

KAA designed and conducted all the experiments, collected and analyzed all the data, and wrote the first draft of the
manuscript. BAM advised the chemical ecology work and revised the manuscript. SMP advised the sensory and nutritional
ecology work and revised the manuscript. CJW secured funding, supervised all of the experimental work, and revised the
manuscript.

**Conflict of interests**

Authors declare no conflict of interests.

**Data availability**

Datasets analyzed and presented in this study are available from the corresponding author upon request. Data sets will be
uploaded to a data repository and this statement will be updated once data are published and an accession number or DOI
is provided.

**References**

[revised manuscript text omitted]

ND: not detected; * Indicates significant quantitative differences between volatiles of weed species as measured by peak area.

700
701
702

Table 2. Treatment list for sensory treatments carried out on the carabid species under study with associated treatment descriptions.

Treatment number and code	Treatment description	
Positive control (+/+)	Carabid beetles with fully functional sensory organs (intact carabid beetles)	Seeds of Brassica napus L. Seeds of Sinapis arvensis L. Seeds of Thlaspi arvense L.
Unilateral control (+/-)	Carabid beetles with fully functional sensory organs on only one side of the body	Seeds of Brassica napus Seeds of Sinapis arvensis Seeds of Thlaspi arvense
Negative control (-/-)	Carabid beetles with all sensory organs blocked and removed (no sensory perception)	Seeds of Brassica napus Seeds of Sinapis arvensis Seeds of Thlaspi arvense
(Antennae + Palps)	Carabid beetles with functional antennae and palps	Seeds of Brassica napus Seeds of Sinapis arvensis Seeds of Thlaspi arvense
(Antennae)	Carabid beetles with functional antennae	Seeds of Brassica napus Seeds of Sinapis arvensis Seeds of Thlaspi arvense
(Palps)	Carabid beetles with functional maxillary and labial palps	Seeds of Brassica napus Seeds of Sinapis arvensis Seeds of Thlaspi arvense
(Maxillary Palps)	Carabid beetles with functional maxillary palps	Seeds of Brassica napus Seeds of Sinapis arvensis Seeds of Thlaspi arvense
(Labial Palps)	Carabid beetles with functional labial palps	Seeds of Brassica napus Seeds of Sinapis arvensis Seeds of Thlaspi arvense
(Eyes)	Carabid beetles with functional compound eyes and ocelli	Seeds of Brassica napus Seeds of Sinapis arvensis Seeds of Thlaspi arvense

**Figure 1.** Seed detection success of sensory-manipulated *Poecilus corvus* (a), *Pterostichus melanarius* (b), and *Amara*
*littoralis* (c) beetles as measured by total number of seeds consumed (mean total seed consumption \pm mean standard error).
Seed choice responses of sensory-manipulated *Poecilus corvus* (d), *Pterostichus melanarius* (e), and *Amara littoralis* (f)
beetles as measured by numbers of seeds consumed from each species (mean number of seeds consumed \pm mean standard
error). (+/+): positive control (intact insects); (+/-): unilateral control (half sensory capability); (-/-): negative control (zero
sensory capability).

**Figure 2.** Seed volatile chemical compounds detected in surface seed extracts of *Brassica napus* (a), *Sinapis arvensis* (b),
and *Thlaspi arvense* (c) measured as total ion content TIC (ion abundance) in *mV*. Numbers represent compounds: (1)
Nonanal, (2) n-Tetradecanoic acid, (3) Hexadecanoic acid ethyl ester, (4) E-9-Octadecanoic acid ethyl ester, (5)
Hexacosane, (6) Hepatacosane, (7) Nonacosane, (8) 15-Nonacosanone.

**Figure 3.** Feeding responses (mean number of seeds consumed \pm mean standard error) of three carabid species offered
seeds of *B. napus* coated with the surface chemicals of *T. arvense* against uncoated intact *B. napus* seeds (a), and seeds of
*T. arvense* coated with the surface chemicals of *B. napus* seeds against uncoated *T. arvense* (b) in two-choice feeding
bioassays.
